# RCCDA: Adaptive Model Updates in the Presence of Concept Drift under a Constrained Resource Budget

**Adam Piaseczny**
Purdue University
apiasecz@purdue.edu

**Md Kamran Chowdhury Shisher**
Purdue University
mshisher@purdue.edu

**Shiqiang Wang**
IBM Research
shiqiang.wang@ieee.org

**Christopher G. Brinton**
Purdue University
cgb@purdue.edu

## Abstract

Machine learning (ML) algorithms deployed in real-world environments are often faced with the challenge of adapting models to concept drift, where the task data distributions are shifting over time. The problem becomes even more difficult when model performance must be maintained under adherence to strict resource constraints. Existing solutions often depend on drift-detection methods that produce high computational overhead for resource-constrained environments, and fail to provide strict guarantees on resource usage or theoretical performance assurances. To address these shortcomings, we propose RCCDA: a dynamic model update policy that optimizes ML training dynamics while ensuring compliance to predefined resource constraints, utilizing only past loss information and a tunable drift threshold. In developing our policy, we analytically characterize the evolution of model loss under concept drift with arbitrary training update decisions. Integrating these results into a Lyapunov drift-plus-penalty framework produces a lightweight greedy-optimal policy that provably limits update frequency and cost. Experimental results on four domain generalization datasets demonstrate that our policy outperforms baseline methods in inference accuracy while adhering to strict resource constraints under several schedules of concept drift, making our solution uniquely suited for real-time ML deployments.

## 1 Introduction

In recent years, deep learning (DL) models have become popular in resource-constrained environments [1, 2, 3, 4], e.g., on mobile devices, where computational power and memory are limited. These settings typically assume static data distributions during inference, in part due to resource limitations [5, 6, 7]. In practice, this poses a major challenge in the form of concept drift, which is a phenomenon where the underlying relationships in data shift over time, causing model performance to degrade [8, 9]. The computational demands of frequent updates that may be necessary to overcome rapid drifts present a significant challenge in resource-constrained settings. As a result, effectively managing model updates to adapt to such drift while adhering to resource constraints becomes a critical problem.

To address concept drift and mitigate model performance deterioration, researchers have proposed a variety of adaptive learning methods [10, 11, 12]. These strategies are often tailored to different operational contexts and environment characteristics [13, 14, 15, 16]. While most are heuristic, validating their efficacy empirically [17, 18], certain techniques have offered convergence guarantees to give principled insights into algorithm design, e.g., [19, 20]. Even so, these theoretical analyses

39th Conference on Neural Information Processing Systems (NeurIPS 2025).

often rely on specific assumptions about the drift, or require prior knowledge about the environment that may not be readily available for decision-making.

**Fundamental Challenges.** The question of how to maintain strong performance in the presence of concept drift under strict resource constraints remains largely unexplored. Existing lightweight adaptation algorithms do not provably guarantee resource budgets will be met during drift adaptation. On the other hand, established convergence guarantees depend on unconstrained updates: as we will see, artificially restricting these algorithms to stay within the budget leads to suboptimal performance. Thus, the fundamental challenge lies in developing a theoretically grounded scheme that guarantees adherence to strict resource constraints while optimizing model performance — a long-standing open problem in the field. To the best of our knowledge, no prior work has proposed or systematically analyzed strategies that optimally address concept drift while prioritizing rigid resource constraints. As a result, in this paper, we are motivated to answer the following question:

*How can we develop a lightweight, resource-aware policy for dynamic model updates that (i) effectively mitigates performance loss incurred by concept drift in real-time applications while (ii) theoretically guaranteeing adherence to strict resource constraints?*

**Contributions.** Through our investigation, we develop RCCDA (Resource-Constrained Concept Drift Adaptation): a first-of-its-kind adaptive threshold-based policy that determines the optimal time to update model parameters in the presence of concept drift, using only past loss information, and ensuring strict bounds on constraint violations. Our contributions are summarized as follows:

- **Convergence Analysis**: We provide a rigorous analysis of how concept drift impacts a deployed model's convergence over time, focusing on the evolution of loss as data distributions shift. Under mild assumptions, our bounds quantify the degradation caused by concept drift, implicitly relating model performance to the retraining policy choice. This analysis lays the groundwork for development of adaptive strategies in dynamic, resource-constrained settings.

- **Optimal Policy Design**: We formulate and solve an optimization problem to determine the optimal times for updating a model, minimizing the loss induced by concept drift while satisfying resource constraints. Leveraging the Lyapunov drift-plus-penalty framework [21], we develop RCCDA: a threshold-based policy that triggers updates based on inference loss data and a virtual queue tracking resource expenditure. The resulting lightweight algorithm relies solely on historical loss data and a tunable drift threshold, making it efficient for real-time applications, while ensuring a greedy-optimal balance between model performance and computational overhead.

- **Empirical Validation**: We evaluate RCCDA performance through experiments on four domain generalization datasets: PACS, DigitsDG, OfficeHome, and MEMD-ABSA, simulating different concept drift schedules. Our results demonstrate that our policy consistently outperforms four baseline policies across a variety of drift settings while adhering to resource budgets. This highlights our policy's ability to adapt efficiently to data with time-varying distributions.

## 2 Related Work

**Concept Drift.** Many of the existing works on studying concept drift have placed an emphasis on characterizing or detecting drift [22, 23, 24, 25, 26]. For instance, windowing techniques, such as the Adaptive Windowing (ADWIN) algorithm [27], dynamically adjust the size of a data window to monitor statistical changes, flagging drift when significant deviations occur. Alternatively, observing performance indicators [28, 29, 30, 31, 32] involves tracking metrics like accuracy or error rates, where a sudden decline may signal that the model no longer aligns with the current data distribution. Additionally, machine learning models can be trained to detect drift by identifying shifts in feature distributions or relationships among variables [33, 34]. Detecting drift is a crucial step in drift adaptation, as it provides actionable insights into the optimal times to update or retrain the model with fresh data [19, 35]. However, these methods do not consider operating under limited resource budget, a problem that is addressed by the policy introduced in our paper.

**Domain Generalization.** Domain generalization (DG) techniques address the related problem of training models to generalize well across multiple domains, aiming to improve robustness to unseen target distributions during inference [36, 37, 38, 39]. Classical strategies include data/feature augmentation [40, 41, 42] or leveraging domain-invariant features [43, 44, 45]. More recent works have investigated temporal domain generalization [46, 47, 48] to adapt to time-varying distributions,

as encountered during concept drift. In practice, however, these approaches demand significant resources, such as continuous data monitoring and frequent model updates, making them impractical for resource-constrained settings. Furthermore, these works have not explicitly considered adherence to resource constraints, which is one of the key motivations of our paper.

**Continual Learning.** As mitigating concept drift involves retraining the model at strategic intervals guided by a policy, the drift-adaptation problem shares similarities with continual learning (CL), where the goal is to enable models to learn from a sequence of tasks over time [49, 50]. A major hurdle in CL is the problem of catastrophic forgetting, where adapting to new tasks causes the model to lose performance on previously learned ones [51, 52]. The problem studied in this paper prioritizes a different objective than CL: optimizing model performance on the *current* data distribution at each time step under resource constraints, i.e., immediate adaptability versus long-term memory retention. As such, our work is applicable to settings where the concept drift does not exhibit cyclic patterns.

## 3 Methodology

### 3.1 System Model

We consider a system with an agent running an inference task using a machine learning model within an environment that evolves over discrete time $t \in \{0, 1, \ldots, T-1\} := \mathcal{T}$. The environment is characterized by an underlying data distribution $p_t$ that changes over time due to concept drift. Consequently, the agent's available dataset, $\mathcal{D}_t \sim p_t$, varies at each time step, resulting in changing conditional distributions of the data:

$$\Pr(y_t \mid x_t) \neq \Pr(y_{t'} \mid x_{t'}), \tag{1}$$

where $(x_t, y_t) \in \mathcal{D}_t$ and $(x_{t'}, y_{t'}) \in \mathcal{D}_{t'}$ are feature-target pairs sampled at different times $t$ and $t'$. We assume that the concept drift experienced by $p_t$ is bounded between any time steps $t$ and $t+1$, which is expressed by the time-varying bound $\delta(t)$ on natural-log-based KL-divergence between consecutive data distributions:

$$\mathrm{D_{KL}}(\mathcal{D}_t \mid\mid \mathcal{D}_{t+1}) \leq \delta(t) \ \forall t, \delta(t) > 0. \tag{2}$$

At any time $t$, the agent uses its current model, with parameters $\theta_t \in \mathbb{R}^d$, to perform the inference task. The inference performance is measured using a local loss function, $f : \mathbb{R}^d \times \mathbb{R}^{|\mathcal{D}_t| \times |x|} \to \mathbb{R}$, the definition of which remains fixed throughout the entire operation. By randomly sampling an inference batch $\xi_t$ (where $|\xi_t| \leq |\mathcal{D}_t|$) from $\mathcal{D}_t$, the agent obtains an unbiased estimate of the loss, $f(\theta_{t-1}, \xi_t)$ (s.t. $\mathbb{E}_{\xi_t \sim \mathcal{D}_t}[f(\theta_{t-1}, \xi_t)] = f(\theta_{t-1}, \mathcal{D}_t)$, and $\theta_{-1} = \theta_0$). Before the agent is deployed, the agent's model (with parameters $\theta_0$) is pretrained to be within some $\varepsilon_0 > 0$ proximity to the initial optimal model, i.e. $\|\theta_0 - \theta_0^*\| \leq \varepsilon_0$ for some $\varepsilon_0 > 0$, where $\theta_0^* := \arg\min_{\theta \in \mathbb{R}^d} f(\theta, \mathcal{D}_0)$, ensuring a low initial inference loss estimate value.

Due to the concept drift experienced by the underlying distribution $p_t$, the optimal model parameters, $\theta_t^* := \arg\min_\theta f(\theta, \mathcal{D}_t)$, also change over time, resulting in the model experiencing a growing inference loss and performance degradation. This problem is illustrated in Figure 1a.

To address this issue, the agent performs actions aimed at keeping the loss increase to a minimum in the presence of concept drift. To achieve that, the agent follows a model update policy, where at any time $t$, the model parameters $\theta_t$ can be updated with the data from the most recent dataset $\mathcal{D}_t$, with the number of updates constrained by the resources available. A single update is formulated as:

$$\theta_t = \theta_{t-1} - \eta \nabla_\theta f(\theta_{t-1}, \zeta_t), \tag{3}$$

where $\eta > 0$ is the learning rate and $\zeta_t \subseteq \mathcal{D}_t \sim p_t$ is a uniformly sampled training batch of data (independent of the $\xi_t$ inference batch). The choice of policy affects the model performance over time, as illustrated in Figure 1b.

Note that while the update in Equation (3) is formulated as a single gradient descent step, this framework can be easily extended to allow for multiple update rounds per time step. This is achieved by interpreting each gradient descent step as a sub-interval of a larger decision time frame, allowing for more granular control over the effective update frequency.

Finally, each model update is associated with a time-varying resource cost $\lambda(t) > 0$. This cost is an abstract measure and can represent a variety of factors, such as computational load, energy

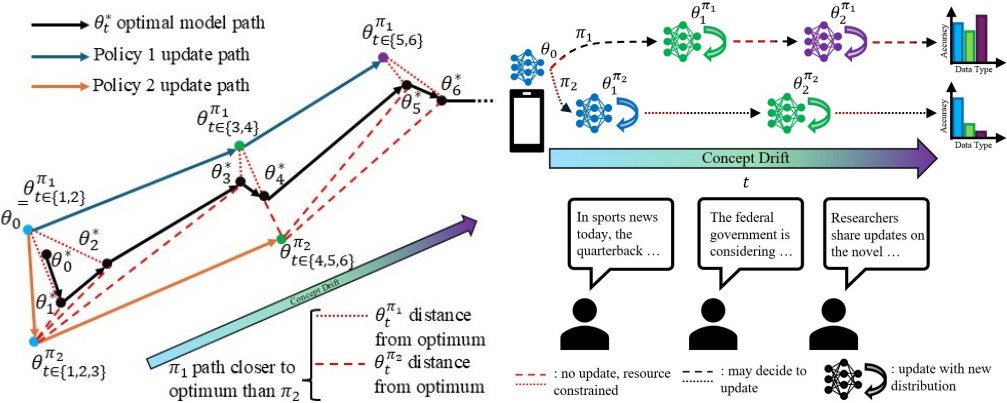

(a) Optimal point and model parameter drift    (b) Effect of different policies on model performance

Figure 1: Interplay between policy design, concept drift, and model performance. **(a)** Optimal model point movement over time due to drift. Different policies result in varying total distances to the drifting optimal model. **(b)** Update timing influences average performance given set number of updates.

consumption, wall-clock time, or monetary cost (details available in Appendix D). The cost relates the effective frequency of updates to the pre-defined time-average cost budget $\bar{\lambda}$, which corresponds to the maximum average resources the agent can spend per time step over the entire horizon $\mathcal{T}$.

### 3.2 Problem Formulation

To mitigate concept drift, the agent employs a model retraining policy. At any time $t$, the policy decides on an action $\pi(t) \in \{0, 1\}$, where $\pi(t) = 1$ and $\pi(t) = 0$ correspond to a model update and no update respectively. The resulting sequence of actions over the entire horizon is denoted by the vector $\pi = (\pi(0), \dots, \pi(T-1))$. This sequence is an element of the action sequence space $\Pi = \{0, 1\}^T$. Utilizing a retraining policy results in the following evolution of the model parameters over time:

$$\theta_t = \theta_{t-1} - \pi(t)\eta\nabla_\theta f\left(\theta_{t-1}, \zeta_t\right). \tag{4}$$

For the sake of practicality and computational efficiency of our solution, we consider policies that are causal and distribution ignorant, meaning (i) the decision maker neither evaluates nor knows the data distributions $p_0, p_1, \dots p_{T-1}$ and (ii) the decision maker decides on the action at time $t$ based on the history of expected losses of the past inference batches of data $\xi_0, \xi_1, \dots, \xi_t$, which are available through running the inference task. This is an important distinction of our setting, as any further modifications to the action space would result in an increased task overhead. For example, trying to detect the concept drift requires performing operations on the dataset, which is highly resource consuming (see Appendix D for more details). Using the inference loss to evaluate the model performance is a common established strategy [53].

Given such a setup, the goal of the agent is to find a policy resulting in a set of actions $\pi \in \Pi$ that minimizes the expected loss over the entire time horizon, subject to the average cost of model updates not exceeding $\bar{\lambda}$:

$$\min_{\pi \in \Pi} \frac{1}{T} \sum_{t=0}^{T-1} \mathbb{E}_\pi\left[f(\theta_t, \mathcal{D}_t)\right] \qquad \text{subject to } \frac{1}{T} \sum_{t=0}^{T-1} \lambda(t)\pi(t) \leq \bar{\lambda}. \tag{5}$$

**Optimality.** Without resource constraints, the optimal solution to (5) would be to update the model at every time $t$. However, finding an optimal update policy under the resource constraint presents a fundamental challenge: obtaining such a policy requires knowledge of the dataset's entire future evolution, which is not practical. To address this, we pivot from the direct time-averaged loss objective in (5) to a new objective based on the provable upper bound on the time-averaged gradient norm (formally derived in Theorem 5.1). This pivot allows us to naturally decompose the problem into a sum of per-time-step performance penalties. In the following section, we introduce the Lyapunov drift-plus-penalty framework, which provides a mechanism to greedily minimize the upper bound on these per-time-step penalties while provably upholding the long-term resource constraint.

# 4 Policy Design

**Idea and Overview.** The central challenge of this work is adapting to changing conditions while respecting resource constraints. This issue spans beyond machine learning, appearing in areas where responsiveness and resource management must align, and requires strategies that balance performance and efficiency across dynamic systems. Existing solutions often rely on computationally expensive methods or fail to provide strict guarantees on resource usage and performance. To overcome these limitations, we leverage the Lyapunov drift-plus-penalty framework [21], as it allows us to transform an intractable long-term optimization problem into a series of greedy, per-time-step decisions. The core intuition is to dynamically incorporate the long-term constraint into each per-time-step objective. This is achieved by defining a "penalty" term, which represents our immediate objective (based on minimizing the convergence bound), and a "drift" term, which measures how far the system deviates from its long-term resource budget. By optimizing a bound on this combined drift-plus-penalty expression at each time $t$, the agent is compelled to make greedy, online decisions that provably balance these two competing factors. This process creates an explicit, online trade-off between short-term performance and long-term budget adherence, allowing us to derive a principled, lightweight policy with provable guarantees.

By tailoring the Lyapunov framework to the analyzed setting, we derive RCCDA (Algorithm 1), a novel policy that dynamically determines when to update the model based on historical performance metrics, while adhering to resource constraints. The algorithm relies on a threshold-based rule, triggering an update when a constructed expression exceeds a dynamic threshold influenced by past decisions and constraints, reflecting the performance-resource trade-off. This approach offers distinct benefits over traditional methods: it avoids complex drift detection, requires few assumptions about data behavior, is simple to implement, and provably guarantees adherence to resource constraints.

## 4.1 Policy Derivation

**Lyapunov Analysis.** To develop RCCDA, we apply the Lyapunov framework to the per-step penalty objective derived from our convergence bound (Theorem 5.1), subject to the resource constraint. The time-averaged sum of these per-step penalties defines our core objective:

---

**Algorithm 1:** RCCDA

**Input:** model parameters $\theta_0$, update cost $\lambda(t)$, time-average cost $\bar{\lambda}$, tunable weight $V$, learning rate $\eta$, estimator $\hat{\mathcal{G}}$

**Output:** Model parameters $\theta_t \forall t$

1   Virtual Queue $Q(0) \leftarrow 0$
2   History Variables $\mathcal{H}_f, \mathcal{H}_g, \mathcal{H}_\pi \leftarrow \{\}, \{\}, \{\}$
3   **for** $t = 0$ **to** $T - 1$ **do**
4     $f_t \leftarrow \mathbb{E}_{\xi_t \sim \mathcal{D}_t} [f(\theta_{t-1}, \xi_t)]$
5     $\mathcal{H}_f \leftarrow \mathcal{H}_f \cup \{f_t\}$
6     $\hat{\mathcal{G}}_t \leftarrow \hat{\mathcal{G}}(\mathcal{H}_f, \mathcal{H}_g, \mathcal{H}_\pi)$
7     $\mathcal{C}(t) \leftarrow \lambda(t)Q(t) + \frac{1}{2}\left[\lambda(t)^2 - 2\bar{\lambda}\lambda(t)\right]$
8     **if** $V\hat{\mathcal{G}}_t \geq \mathcal{C}(t)$ **then**
9       $\pi(t) \leftarrow 1$
10      $\theta_t \leftarrow \theta_{t-1} - \eta\nabla f(\theta_{t-1}, \zeta_t)$
11      $\mathcal{H}_g \leftarrow \mathcal{H}_g \cup \{\nabla f(\theta_{t-1}, \zeta_t)\}$
12     **else**
13      $\pi(t) \leftarrow 0$
14      $\theta_t \leftarrow \theta_{t-1}$
15     $\mathcal{H}_\pi \leftarrow \mathcal{H}_\pi \cup \{\pi(t)\}$
16     $Q(t+1) \leftarrow \max\{0, Q(t) + \lambda(t)\pi(t) - \bar{\lambda}\}$

---

$$\frac{1}{T}\sum_{t=0}^{T-1}\left(\underbrace{f(\theta_t, \xi_t) - f(\theta_{t+1}, \xi_{t+1})}_{\text{model related loss difference}} + (1 - \pi(t))\underbrace{\Delta f_{\delta,\xi}(t)}_{\text{drift loss}}\right), \tag{6}$$

where for every time step $t$, $f(\theta_t, \xi_t) - f(\theta_{t+1}, \xi_{t+1})$ is an inference loss term associated with the evolution of model parameters over time, which itself depends on the dataset evolution too, and $\Delta f_{\delta,\xi}(t) := f(\theta_t, \xi_{t+1}) - f(\theta_t, \xi_t)$ is the drift induced inference loss term that relates the evolution of the data stream with the incurred loss. Note that this objective is derived from the right-hand side of our convergence bound in Theorem 5.1, which will be presented in Section 5, and serves as a proxy for minimizing the bound on time-averaged gradients, $\frac{1}{T}\sum_{t=0}^{T-1}\|\nabla f(\theta_t, \mathcal{D}_{t+1})\|^2$.

The practical objective in (6) introduces two key modifications from the theoretical bound. First, it explicitly incorporates the policy decision by scaling the drift loss with $1 - \pi(t)$, isolating the drift penalty to non-update steps. Second, while the theoretical bound is defined for losses over the entire (and often inaccessible) datasets $\mathcal{D}_t$, our objective employs the practical losses computed on inference batches $\xi_t$ for different times $t$.

**Virtual Queue.** To enforce adherence to resource constraints over time, we introduce a virtual queue $Q(t)$, which evolves over time according to:

$$Q(t+1) = \max\left\{0, Q(t) + \lambda(t)\pi(t) - \bar{\lambda}\right\}, \tag{7}$$

with $Q(0) = 0$. Intuitively, the virtual queue acts as a "debt accumulator," which tracks the violations of the resource constraint over time. When an update occurs, $\pi(t) = 1$, the queue grows in size by $\lambda(t)$, signaling resource expenditure; the subtraction of $\bar{\lambda}$ reflects the permissible usage rate. As a result, the queue evolves dynamically over time, signifying excessive updates when large, or suggesting room for more updates when close to $0$. This mechanism is central to the Lyapunov framework: by seeking to keep the queue stable (i.e., minimizing its "drift"), the resulting optimization ensures the average resource consumption aligns with $\bar{\lambda}$ over time.

To formalize this, we first define the Lyapunov drift as $\Delta Q(t) := \frac{1}{2}\left(Q(t+1)^2 - Q(t)^2\right)$, which corresponds to change in the "resource debt". The core of the method is to greedily minimize a bound on the single drift-plus-penalty expression at each time $t$, which is a weighted sum of this drift and our performance objective:

$$\mathbb{E}[\Delta Q(t)] + V\Psi(t), \tag{8}$$

where $\Psi(t)$ is the per-step penalty objective (the term inside the summation in Equation 6), and $V > 0$ is a tunable parameter that weights the penalty term against the Lyapunov drift.

## 4.2 Our Algorithm

**Decision-Making.** Solving the optimization problem involves greedily minimizing the bound on the drift-plus-penalty expression at each time step $t$. The result of the derivation, available in Appendix C, is a theoretically greedily-optimal threshold-based policy at the core of RCCDA (Algorithm 1) that dictates an update ($\pi(t) = 1$) if and only the following condition holds:

$$V\left(f(\theta_{t-1}, \xi_{t+1}) - f(\theta_{t-1} - \eta\nabla f(\theta_{t-1}, \zeta_t), \xi_t)\right) + V\mathcal{J}_\pi(t) \geq \lambda(t)Q(t) + \frac{1}{2}\left[\lambda(t)^2 - 2\bar{\lambda}\lambda(t)\right], \tag{9}$$

where $\mathcal{J}_\pi(t) := f(\theta_{t-1} - \eta\nabla f(\theta_{t-1}, \zeta_t) - \pi(t+1)\eta\nabla f(\theta_{t-1} - \eta\nabla f(\theta_{t-1}, \zeta_t), \zeta_{t+1}), \xi_{t+1}) - f(\theta_{t-1} - \pi(t+1)\eta\nabla f(\theta_{t-1}, \zeta_{t+1}), \xi_{t+1})$. The intuition behind the update rule is a direct trade-off. The left-hand side, which we denote $V\mathcal{G}(t)$, represents the performance gain scaled by the trade-off parameter $V$. The $\mathcal{G}(t)$ term is a complex, non-causal expression that accounts for the loss reduction from the gradient step and the nested effects of how this action impacts the loss at time $t+1$, given the optimal next action $\pi(t+1)$. This scaled gain is compared against the right-hand side, denoted $\mathcal{C}(t)$, which represents the effective resource cost. This cost term is fully known at time $t$ and combines the cost associated with the current resource debt, $\lambda(t)Q(t)$, with the instantaneous cost of the update itself. The resulting rule $V\mathcal{G}(t) \geq \mathcal{C}(t)$, dictates that the agent should update only when the scaled performance benefit is greater than the resource cost.

**Practical Policy and Estimator.** While the threshold in (9) provides a per-time-step optimal solution, it is intractable and non-causal, making it impossible to implement in practice. The performance gain term $\mathcal{G}(t)$ directly depends on the future data distribution $\mathcal{D}_{t+1}$, the optimal future action $\pi(t+1)$ (which itself creates a recursive dependency), and computations of various gradient terms that cannot be performed without violating resource constraints. There are several strategies to address this intractability, such as optimizing a looser convergence bound or truncating the multi-step dependencies. In this work, however, we retain the structure of the ideal policy and instead utilize a causal approximation of this rule. We define the practical policy for RCCDA by replacing the intractable gain $\mathcal{G}(t)$ with a causal estimator function, $\hat{\mathcal{G}}(\mathcal{H}_t)$, which utilizes only historical information $\mathcal{H}_t = \{\mathcal{H}_{f_{0:t}}, \mathcal{H}_{g_{0:t}}, \mathcal{H}_{\pi_{0:t}}\}$ available up to time $t$. The practical decision rule thus becomes $V\hat{\mathcal{G}}(\mathcal{H}_t) \geq \mathcal{C}(t)$. While this design introduces an implicit estimation error that may affect performance, it also establishes RCCDA as a flexible framework, allowing the specific choice of the estimator $\hat{\mathcal{G}}$ to be tailored to different problems.

**RCCDA Operation.** Based on this practical, estimator-based policy, the RCCDA algorithm (Algorithm 1) operates over $\mathcal{T} = \{0, \ldots, T-1\}$. After initializing the virtual queue $Q(0) = 0$ and history variables $\mathcal{H}$, the algorithm proceeds at each time step $t$ by first computing the current inference loss $f_t$ and updating the loss history $\mathcal{H}_f$. It then uses its chosen estimator function $\hat{\mathcal{G}}(\mathcal{H}_t)$ to calculate

the estimated scaled performance gain $V\hat{\mathcal{G}}(\mathcal{H}_t)$. This gain is compared to the known resource cost threshold $\mathcal{C}(t)$ (the RHS of (9)) to make the update decision $\pi(t)$. Based on this action, the model parameters $\theta_t$ are either updated or kept constant, and the relevant histories are recorded. Finally, the virtual queue is updated to $Q(t+1)$ using the action taken, $\pi(t)$, and its cost $\lambda(t)$, carrying the resource debt forward to the next time step.

## 5    Theoretical Analysis

In this section we conduct a theoretical analysis of the examined setting. In the analysis, we consider a perfect estimator, i.e. $\hat{\mathcal{G}}(\mathcal{H}_t) = \mathcal{G}(t)$, which achieves zero estimation error. This allows us to derive performance bounds for the ideal policy, which serve as a theoretical benchmark for any practical implementation. The analysis is structured as follows: We first introduce a set of common assumptions required for the analysis. We then derive the general convergence rate bounds and conduct and demonstrate the result of a stability analysis on the proposed update policy. The complete proofs of our theorems are available in the appendix.

### 5.1    Convergence Analysis

**Assumption 1 (Smoothness).** *The agent's loss function is differentiable and L-smooth, i.e. $\forall t_0, t_1, t_2$ it follows that $\|\nabla f(\theta_{t_1}, \mathcal{D}_{t_0}) - \nabla f(\theta_{t_2}, \mathcal{D}_{t_0})\| \leq L \|\theta_{t_1} - \theta_{t_2}\|$.*

**Assumption 2 (Bounded Variance).** *For all times $t$, the agent's stochastic gradient $\nabla f(\theta_t, \zeta_t)$ is an unbiased estimate of the true gradient, i.e. $\mathbb{E}_{\zeta_t \sim \mathcal{D}_t} [\nabla f(\theta_t, \zeta_t)] = \nabla f(\theta_t, \mathcal{D}_t)$, and the variance of the stochastic gradient is bounded as $\mathbb{E}_{\zeta_t \sim \mathcal{D}_t} \left[ \|\nabla f(\theta_t, \zeta_t) - \nabla f(\theta_t, \mathcal{D}_t)\|^2 \right] \leq \sigma^2$.*

**Assumption 3 (Bounded Concept Drift).** *For all times $t$, the magnitude of the concept drift between two data distributions $p_t$ and $p_{t+1}$ is bounded, which corresponds to a bound on the KL-divergence measure on datasets at two consecutive times defined as $\forall t, \exists \delta(t) > 0$ s.t. $D_{\mathrm{KL}}(\mathcal{D}_t \parallel \mathcal{D}_{t+1}) \leq \delta(t) \leq \delta = \sup \{\delta(0), \ldots, \delta(T)\}$.*

**Assumption 4 (Bounded Loss).** *For all times $t$, the magnitude of the loss $f(\theta_t, \mathcal{D}_t)$ satisfies $0 \leq f(\theta_t, \mathcal{D}_t) \leq B$.*

**Theorem 5.1.** *If Assumptions 1-2 hold and the learning rate is chosen such that $\eta < \frac{2P_{\min}^\pi}{L}$, then the time-averaged gradient satisfies*

$$\frac{1}{T} \sum_{t=0}^{T-1} \|\nabla f(\theta_t, \mathcal{D}_{t+1})\|^2 \leq \frac{1}{T\mu} \left( f(\theta_0, \mathcal{D}_0) - f(\theta_T, \mathcal{D}_T) + \sum_{t=0}^{T-1} \Delta f_\delta(t) \right) + \frac{L\eta}{2P_{\min}^\pi - L\eta} \sigma^2, \tag{10}$$

*where $T$ is the total number of time steps, $\Delta f_\delta(t) = f(\theta_t, \mathcal{D}_{t+1}) - f(\theta_t, \mathcal{D}_t)$ is the time-varying drift-induced loss change, $\mu = \left( \eta P_{\min}^\pi - \frac{L\eta^2}{2} \right)$, and $P_{\min}^\pi$ is the minimum probability of policy $\pi(t)$ being equal to $1$ for all times $t$.*

**Corollary 5.2.** *Under Assumptions 3-4, by the result in **Theorem 5.1**, the time-averaged gradient satisfies*

$$\frac{1}{T} \sum_{t=0}^{T-1} \|\nabla f(\theta_t, \mathcal{D}_{t+1})\|^2 \leq \frac{B}{T\left(\eta P_{\min}^\pi - \frac{L\eta^2}{2}\right)} \left( 1 + \sqrt{2} \sum_{t=0}^{T-1} \sqrt{\delta(t)} \right) + \frac{L\eta}{2P_{\min}^\pi - L\eta} \sigma^2 \tag{11}$$

There are several key steps in our convergence proofs. By incorporating the drift-induced loss term $\Delta f_\delta(t)$, we are able to quantify the effect of shifting data distributions between time steps. Utilizing Pinsker's inequality allows us to bound $\mathbb{E}[\Delta f_\delta(t)] \leq B\sqrt{2\delta(t)}$, establishing an explicit connection between the drift induced loss and the magnitude of concept drift. Conditioning on the past dataset and model evolution allows us to model dependencies on past states, and decouple the policy action from the evolution of loss over time. Finally, by directly considering the action $\pi(t)$ in the model update and introducing $P_{\min}^\pi$, we are able to integrate general policy information into the final bound. A detailed proof is provided in Appendix B.

**Discussion.** Theorem 5.1 provides a general bound on the time-averaged squared gradient norm, quantifying the impact of concept drift and the update policy on convergence. The bound's dependence on the cumulative drift, $\sum_{t=0}^{T-1} \Delta f_\delta(t)$, reveals an important insight: the time-averaged gradient only vanishes if the total drift grows sublinearly with $T$. Persistent, non-vanishing drift prevents the gradients from converging. The policy's role is twofold: it explicitly appears in $P_{\min}^\pi$, which inversely scales both the drift-dependent term (via $\mu$) and the error term $\mathcal{O}(\sigma^2)$, highlighting the trade-off where more frequent updates (a higher $P_{\min}^\pi$) lead to a tighter convergence bound, at the cost of more computation. Implicitly, the policy also governs the parameter sequence $(\theta_t)_{t=0}^T$, which determines the loss evolution term $f(\theta_0, \mathcal{D}_0) - f(\theta_T, \mathcal{D}_T) + \sum_{t=0}^{T-1} \Delta f_\delta(t)$. A more effective policy will yield a more favorable loss evolution and thus a tighter overall convergence bound.

Corollary 5.2 demonstrates the relationship between convergence and magnitude of the drift. Without any further assumptions on the drift, the gradients are not guaranteed to converge, even when $\sigma = 0$, as it would require an increasing learning rate, which is bounded by $\eta < \frac{2P_{\min}^\pi}{L}$. If Assumption 3 holds as $t \to \infty$, then $\exists \delta = \sup \{\delta(t) \mid t \in \mathbb{N} \cup \{0\}\}$, and the time averaged gradients are bounded by $\mathcal{O}\left(\sqrt{\bar{\delta}}\right) + \mathcal{O}\left(\sigma^2\right)$, suggesting that in environments with persistent but bounded drift, the model achieves a "stable" performance by oscillating within a region proportional to $\sqrt{\bar{\delta}}$ and $\sigma^2$. Given additional assumptions about the concept drift, it is possible to demonstrate convergence. Suppose $\sum_{t=0}^{T-1} \delta(t) \sim \mathcal{O}(T^{1-\alpha})$ with $\alpha > 0$. Then, if we use $\eta = \mathcal{O}(T^{-\beta})$ such that $0 < \beta < \min(\alpha, 1)$, the bound converges as $\frac{1}{T} \sum_{t=0}^{T-1} \mathbb{E}\left[\|\nabla f(\theta_t, \mathcal{D}_{t+1})\|^2\right] \leq \mathcal{O}(T^{\beta-1}) + \mathcal{O}(T^{\beta-\alpha}) + \mathcal{O}(T^{-\beta})$. In practical scenarios, this corresponds to a decaying drift that eventually vanishes, allowing the algorithm to successfully converge to a stationary point.

## 5.2 Stability Analysis

Now, we discuss how RCCDA 1 satisfies the resource constraint of the problem (5) by following the threshold structure (9). In Theorem 5.3, we provide an upper bound on the constraint violation of our algorithm. This upper bound goes to zero as $T \to \infty$.

**Theorem 5.3.** *If Assumptions 1-4 hold, then the policy following Algorithm 1, with zero-error estimator, satisfies*

$$\frac{1}{T} \sum_{t=0}^{T-1} \lambda(t) \mathbb{E}\left[\pi(t)\right] - \bar{\lambda} \leq \sqrt{\frac{\bar{\lambda}}{T^2} \sum_{t=0}^{T-1} \lambda(t) - \frac{\bar{\lambda}^2}{T} + 2VB\left(\frac{5}{T} + \frac{1}{T^2} \sum_{t=0}^{T-1} \sqrt{2\delta(t)}\right)}. \quad (12)$$

The proof of Theorem 5.3 proceeds in two key steps. First, we show that the constraint violation term (the left hand-side of (12)) is upper bounded by the queue length $Q(T)$, for which an upper bound is also provided. Second, we establish an upper bound for the drift term $\Delta Q(t) = 1/2(Q(t+1)^2 - Q(t)^2)$. This bound is derived by comparing our policy with a stationary random policy that updates at each time slot with probability $\min\left(\bar{\lambda}/\lambda(t), 1\right)$. A detailed proof can be found in Appendix C.2.

**Discussion.** Theorem 5.3 guarantees that RCCDA satisfies the resource constraint in (5) as $T \to \infty$, provided the cumulative drift $\sum_{t=0}^{T-1} \sqrt{\delta(t)}$ grows sublinearly in $T$ and the update costs $\lambda(t)$ remain bounded, i.e., $\lambda(t) \sim \mathcal{O}(1)$. This reflects a key strength of the Lyapunov drift-plus-penalty framework [21]. For finite $T$, the violation bound can be tightened toward zero by tuning V optimally. Additionally, under high but bounded drift rates ($\delta(t) \sim \mathcal{O}(1)$), the constraint violation decays as $\mathcal{O}(1/\sqrt{T})$, ensuring the policy adheres to the budget over long horizons without requiring drift to vanish.

## 6 Experiments

**Datasets.** To represent concept drift, we utilize four widely used domain generalization datasets: PACS [54], DigitsDG [55], and in the appendix OfficeHome [56], and MEMD-ABSA [57]. The drift is simulated by introducing samples from different domains into the dataset. The model architectures utilized are detailed in Appendix E.2. Initially, we pretrain the models on data from a single source domain.

Table 1: Average validation accuracy of policies across concept drift schedules for PACS and DigitsDG datasets, mean update rate constraint: $\frac{\bar{\lambda}}{\lambda} = 0.1$. We see that RCCDA consistently outperforms others, with the largest improvement for Burst and Spikes drift schedules.

| Policy | PACS | | | | Digits-DG | | | |
|---|---|---|---|---|---|---|---|---|
| Drift Schedule | Burst | Step | Wave | Spikes | Burst | Step | Wave | Spikes |
| **RCCDA (ours)** | **72.8** ± 4.5 | **72.0** ± 8.1 | **67.0** ± 4.7 | **73.0** ± 3.6 | **77.6** ± 6.3 | **74.6** ± 7.0 | **65.0** ± 11.2 | **74.3** ± 7.3 |
| Uniform | 64.0 ± 2.2 | 65.3 ± 6.3 | 61.8 ± 7.3 | 60.8 ± 7.7 | 71.4 ± 5.0 | 73.1 ± 4.8 | 63.6 ± 7.5 | 67.0 ± 3.6 |
| Periodic | 65.1 ± 2.4 | 65.5 ± 7.1 | 61.0 ± 7.8 | 55.3 ± 4.0 | 69.8 ± 3.6 | 72.7 ± 4.6 | 63.5 ± 7.5 | 68.1 ± 6.8 |
| Budget-Increase | 59.1 ± 13.5 | 66.2 ± 9.9 | 58.3 ± 9.0 | 53.1 ± 16.4 | 65.9 ± 4.5 | 73.3 ± 5.6 | 63.5 ± 9.2 | 61.9 ± 5.8 |
| Budget-Threshold | 67.4 ± 14.9 | **71.7** ± 8.3 | 54.5 ± 7.0 | 58.4 ± 12.3 | 71.9 ± 3.7 | 72.8 ± 7.5 | 57.4 ± 5.3 | 67.3 ± 6.2 |

**Drift Schedules.** After the pretraining phase, the models are deployed at an agent with some fixed-size dataset, initially comprised only of source domain data. Over time, the dataset experiences concept drift, in the form of influx of data points from different domains, replacing existing data. The incoming domains, drift rate, and time of drift are determined according to a drift schedule. We consider 4 main drift schedules:

(i) *Burst* - at pre-determined times, bursts with high drift rate happen, replacing most of the dataset with new domains. Otherwise, the dataset remains constant.

(ii) *Step* - initially the drift is 0, then at some point, new domains start incoming slowly into the dataset, until the original is introduced back.

(iii) *Wave* - the drift is either 0, or new domain data is introduced at a low rate into the dataset over some long interval periodically.

(iv) *Spikes* - At random times, new domain data is inserted for some time with a random drift rate.

At each time, the agent runs an inference task, calculating inference loss on a separate, smaller holdout dataset that evolves in the same way as the training dataset. The agent has access to a retraining policy that decides when to update the model, as well as the desired time-average-cost $\bar{\lambda}$. Each model update is equivalent to taking $n_{\text{steps}}$ SGD steps, with the gradients computed on batches sampled form the local dataset, and as such we consider a constant update cost $\lambda$. The agent also sets the value of $\bar{\lambda} < \lambda$. The entire framework runs for 250 time steps. The detailed experimental setup is available in Appendix E.

**Policies.** We consider the following update policies.

- RCCDA: Our policy is provided in Algorithm 1. To implement the algorithm, we use a custom estimation function $\hat{\mathcal{G}}(\mathcal{H}_t) = K_p(f(\theta_{t-1}, \xi_t) - \min_{i \in \{0,...,t\}} f(\theta_{i-1}, \xi_i)) + K_d(f(\theta_{t-1}, \xi_t) - f(\theta_{t-1}, \xi_{t-1}))$, where $K_p$ and $K_d$ are tunable constants.

- Uniform Random: The policy updates the model at every time slot with probability $\bar{\lambda}/\lambda$. This policy is motivated from [58].

- Periodic: This deterministic policy updates the model after every time interval $\lambda/\bar{\lambda}$, i.e. $\pi_{\text{Periodic}} = 1$ if $t = k\lceil \lambda/\bar{\lambda} \rceil, k \in \mathbb{N} \cup \{0\}$, and 0 otherwise. The policy is modified from [59].

- Budget-Increase: The policy keeps track of an update budget available over time, then if it detects $n \in \mathbb{N}$ consecutive loss increases and if there is budget available, the model is updated. Based on modified trend-detection methods in [60].

- Budget-Threshold: The policy keeps track of an update budget and a window of losses $\mathcal{W}_f$. Then if $f_t \geq (1 + \epsilon) \max_{f_i \in \mathcal{W}_f} f_i$ and if there is budget available, the model is updated. Based on a modified version of [10].

Given this setup, we evaluate the classification accuracy at all times on the holdout dataset, while keeping track of the number of updates to monitor resource constraint satisfaction. Further experimentation details are available in the supplemenal material section.

**Results and Discussion.** Table 1 presents the average model accuracy under mean update rate constraint of $0.1$. We observe that RCCDA consistently outperforms the baselines across all tested configurations. Furthermore, Figure 2 illustrates the dynamic behavior of accuracy and update rates, confirming that the robust performance is achieved while adhering to the long-term resource constraints.

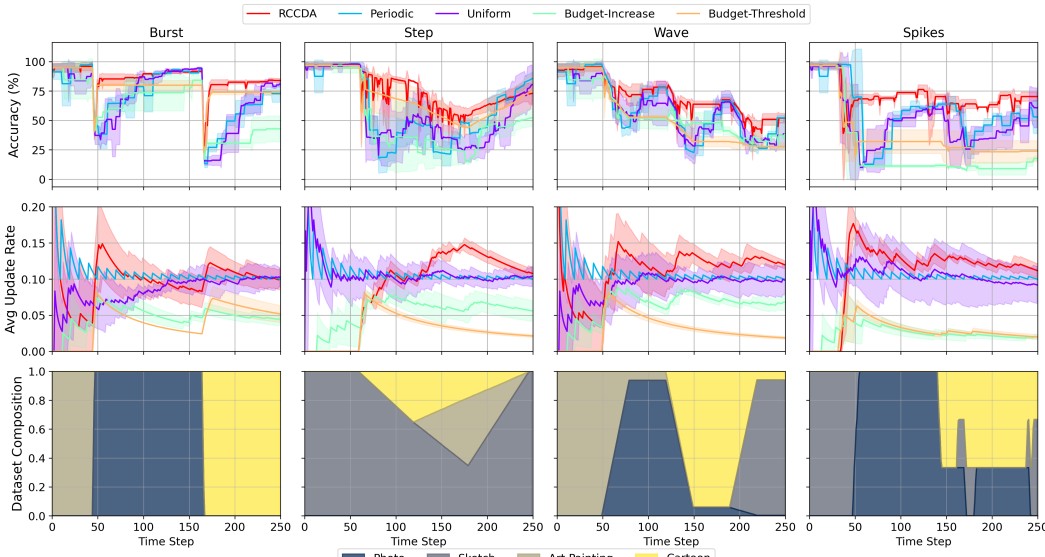

Figure 2: Dynamics of model accuracy and update rate for proposed and baseline policies under various concept drift schedules, with corresponding dataset composition over time. We see that the proposed policy recovers the quickest after sharp transitions while adhering to resource constraints.

**Key Observations.** First, compared with the baselines, RCCDA is able to more effectively adapt to concept drift. Figure 2 demonstrates that for fast drops in accuracy, the update rate increases rapidly, leading to fast performance recovery. This quick adaptation results in performance gains over the baseline polices. Notably, the Budget-Threshold policy also demonstrates a strongly reactive capability, however it cannot adapt to low-drift settings, not updating the model when it could be beneficial. Second, the drift schedule directly affects the performance difference between the policies. For Burst, the rapid domain changes allow the model to quickly adapt to new data, resulting in high average accuracy. RCCDA is able to adapt quickly to these changes, which is why its performance gap is highest on Burst and Spikes. In contrast, the more gradual introduction of data in the Wave and Step schedules contributes to a smaller performance gap. The low accuracy is most likely a result of model capacity limitations, but the small performance difference suggest that at lower drift rates, the time of the update might not be as significant as for higher drift rates.

**Resource Utilization.** Finally, RCCDA demonstrates its update rate converging towards the desired constraint of $0.1$ over time, all while maintaining robust performance. This lends further practical significance to Theorem 5.3, and also suggests that in order to maximize model performance, the policy maximally utilizes the available resources. In contrast, policies focused solely on resource constraints (Uniform, Periodic) fail to update when beneficial for performance, while the budget-based policies are not capable of modeling the complex drift behavior, leading to significant resource under-utilization and lower average accuracy.

**Additional Results and Ablation Studies.** Additional experimental results, including more drift schedules, different update rates, and more datasets, are available in Appendix F.

# 7 Conclusion and Limitations

In this paper, we proposed RCCDA: a novel model update policy for adapting to concept drift under resource constraints. We first established convergence bounds under drift, then utilized Lyapunov analysis to arrive at a policy that minimizes the bound on drift-plus-penalty term per each time slot. We then derived stability bounds, provably demonstrating adherence to resource constraints. Finally, we confirmed the effectiveness of our policy through experiments on various domain generalization datasets, with improvements in accuracy and drift recovery time relative to baselines. A potential limitation of our work is the estimation function $\hat{\mathcal{G}}$ introduced in the algorithm, as its associated error can affect the threshold policy performance. Future work can rigorously characterize this impact and employ it to derive tighter performance bounds.

## Acknowledgements

This paper was supported in part by the National Science Foundation (NSF) under grant CPS-2313109, the Office of Naval Research (ONR) under grant N00014-22-1-2305, and the Air Force Office of Scientific Research (AFOSR) under grant FA9550-24-1-0083.

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

# Appendix

# A   Societal Impact

RCCDA offers significant positive societal contributions by enabling artificial intelligence that is more sustainable, adaptable, and private. By optimizing the efficiency of model updates, our policy directly lowers the energy consumption of edge devices, reducing operational costs, increasing operation times, and promoting environmental sustainability. Additionally, this resource efficiency combined with RCCDA's lightweight nature makes it possible to deploy sophisticated, adaptive models on low-cost hardware, allowing for effective and reliable access to advanced technology even in dynamic, resource-constrained environments. Furthermore, by ensuring this adaptation happens directly on-device, RCCDA provides robust performance while fostering greater data privacy and autonomy for its users.

However, the same adaptive capability that drives these benefits also introduces important considerations for responsible deployment. Since the policy adapts to incoming data, it may inadvertently reinforce harmful patterns, especially if the data stream contains biases. The reliance on incoming data also creates a vulnerability to adversarial poisoning, where a malicious actor could inject data designed to degrade model performance. Finally, continuous on-device personalization could bias the model to favor one type of data, resulting in the creation of insulating "echo chambers," potentially narrowing a user's exposure to different perspectives. As a result, deploying this system responsibly requires integration of safeguards that ensure safe and fair operation, such as bias detection mechanisms, data verification protocols, or regular performance audits.

# B   Convergence Proofs

## B.1   Proof of Theorem 5.1

First, we note that

$$\theta_{t+1} - \theta_t = -\pi(t+1)\eta\nabla_\theta f(\theta_t, \zeta_{t+1}), \tag{13}$$

where $\zeta_{t+1}$ is a random batch sampled uniformly from the dataset $D_{t+1}$ at time $t+1$. For simplicity, in the following analyses, we use a simplified gradient notation $\nabla f(\theta_t, \zeta_{t+1}) := \nabla_\theta f(\theta_t, \zeta_{t+1})$.

The loss difference over two time steps is comprised of two parts:

$$f(\theta_{t+1}, \mathcal{D}_{t+1}) - f(\theta_t, \mathcal{D}_t) = \underbrace{[f(\theta_{t+1}, \mathcal{D}_{t+1}) - f(\theta_t, \mathcal{D}_{t+1})]}_{\text{model-induced loss}} + \underbrace{[f(\theta_t, \mathcal{D}_{t+1}) - f(\theta_t, \mathcal{D}_t)]}_{\Delta f_\delta(t)}. \tag{14}$$

By definition, the second term is the drift-induced loss $\Delta f_\delta(t) := f(\theta_t, \mathcal{D}_{t+1}) - f(\theta_t, \mathcal{D}_t)$.

Next, by **Assumption 1** and Equation 13, the first term $f(\theta_{t+1}, \mathcal{D}_{t+1}) - f(\theta_t, \mathcal{D}_{t+1})$ is bounded as:

$$f(\theta_{t+1}, \mathcal{D}_{t+1}) - f(\theta_t, \mathcal{D}_{t+1}) \le \langle \nabla f(\theta_t, \mathcal{D}_{t+1}), \theta_{t+1} - \theta_t \rangle + \frac{L}{2} \|\theta_{t+1} - \theta_t\|^2. \tag{15}$$

Define the conditional expectation $\mathbb{E}_{t+1}[\cdot] := \mathbb{E}_{\pi(t+1)}[\cdot \mid \mathcal{H}_\theta(t), \mathcal{H}_\mathcal{D}(t+1)]$, where $\mathcal{H}_\theta(t)$ is a variable containing model parameters $\theta_0, \theta_1, \ldots, \theta_t$ across a time horizon $\{0, \ldots, t\}$, and $\mathcal{H}_\mathcal{D}(t)$ stores the evolution of the dataset $\mathcal{D}_0, \mathcal{D}_1, \ldots, \mathcal{D}_{t+1}$ across $\{0, \ldots, t+1\}$. As such, given the history

variables $\mathcal{H}_\theta(t), \mathcal{H}_\mathcal{D}(t+1)$, we obtain:

$$\mathbb{E}_{t+1}[f(\theta_{t+1}, \mathcal{D}_{t+1}) - f(\theta_t, \mathcal{D}_{t+1})]$$

$$\overset{(a)}{\leq} \mathbb{E}_{t+1}\left[\langle \nabla f(\theta_t, \mathcal{D}_{t+1}), \theta_{t+1} - \theta_t\rangle + \frac{L}{2}\|\theta_{t+1} - \theta_t\|^2\right]$$

$$\overset{(b)}{=} \left\langle \nabla f(\theta_t, \mathcal{D}_{t+1}), \mathbb{E}_{t+1}[\theta_{t+1} - \theta_t]\right\rangle + \frac{L}{2}\mathbb{E}_{t+1}\left[\|\theta_{t+1} - \theta_t\|^2\right]$$

$$\overset{(c)}{=} -\left\langle \nabla f(\theta_t, \mathcal{D}_{t+1}), \mathbb{E}_{t+1}[\pi(t+1)\eta\nabla f(\theta_t, \zeta_{t+1})]\right\rangle + \frac{L}{2}\mathbb{E}_{t+1}\left[\eta^2\pi(t+1)^2\|\nabla f(\theta_t, \zeta_{t+1})\|^2\right]$$

$$\overset{(d)}{=} -\left\langle \nabla f(\theta_t, \mathcal{D}_{t+1}), \eta\mathbb{E}_{t+1}[\pi(t+1)]\mathbb{E}_{t+1}[\nabla f(\theta_t, \zeta_{t+1})]\right\rangle$$

$$+ \frac{L\eta^2}{2}\mathbb{E}_{t+1}\left[\pi(t+1)^2\right]\mathbb{E}_{t+1}\left[\|\nabla f(\theta_t, \zeta_{t+1})\|^2\right]$$

$$\overset{(e)}{=} -\eta\mathbb{E}_{t+1}[\pi(t+1)]\left(\left\langle \nabla f(\theta_t, \mathcal{D}_{t+1}), \mathbb{E}_{t+1}[\nabla f(\theta_t, \zeta_{t+1})]\right\rangle - \frac{L\eta}{2}\mathbb{E}_{t+1}\left[\|\nabla f(\theta_t, \zeta_{t+1})\|^2\right]\right)$$

$$\overset{(f)}{=} -\eta\mathbb{E}_{t+1}[\pi(t+1)]\left(\left\langle \nabla f(\theta_t, \mathcal{D}_{t+1}), \nabla f(\theta_t, \mathcal{D}_{t+1})\right\rangle - \frac{L\eta}{2}\mathbb{E}_{t+1}\left[\|\nabla f(\theta_t, \zeta_{t+1})\|^2\right]\right)$$

$$= -\eta\mathbb{E}_{t+1}[\pi(t+1)]\left(\|\nabla f(\theta_t, \mathcal{D}_{t+1})\|^2 - \frac{L\eta}{2}\mathbb{E}_{t+1}\left[\|\nabla f(\theta_t, \zeta_{t+1})\|^2\right]\right),$$

where:

  (a) follows from Equation 15.

  (b) is true due to the fact that $\nabla f(\theta_t, \mathcal{D}_{t+1})$ is $(\mathcal{H}_\theta(t), \mathcal{H}_\mathcal{D}(t+1))$-measurable.

  (c) follows from Equation 13.

  (d) is valid because the policy's action $\pi(t+1)$ is independent of the training data batch $\zeta_{t+1}$ conditioned on the past history.

  (e) is true because $\mathbb{E}_{t+1}[\pi(t+1)] = \mathbb{E}_{t+1}[\pi(t+1)^2]$ since $\pi(t) \in \{0, 1\}$.

  (f) follows from **Assumption 2**.

Next, define $P_{\min}^\pi$ such that $P_{\min}^\pi \leq \mathbb{E}_t[\pi(t)] \ \forall t \in \mathbb{N} \cup \{0\}$ for a given policy $\pi$, which is interpreted as the minimum probability of the policy $\pi$ being equal to 1 at any time $t$, given the evolution of model parameters and datasets up to times $t$ and $t+1$ respectively. Additionally, it is true that $\mathbb{E}_t[\pi(t)] \leq 1 \ \forall t$. Combining this fact with the derived result, it follows that:

$$\mathbb{E}_{t+1}[f(\theta_{t+1}, \mathcal{D}_{t+1}) - f(\theta_t, \mathcal{D}_{t+1})] \leq -\eta P_{\min}^\pi \|\nabla f(\theta_t, \mathcal{D}_{t+1})\|^2 + \frac{L\eta^2}{2}\mathbb{E}_{t+1}\left[\|\nabla f(\theta_t, \zeta_{t+1})\|^2\right]. \tag{16}$$

By applying the law of total expectation to Equation (16), it follows that:

$$\mathbb{E}[f(\theta_{t+1}, \mathcal{D}_{t+1}) - f(\theta_t, \mathcal{D}_{t+1})] \leq -\eta P_{\min}^\pi\mathbb{E}\|\nabla f(\theta_t, \mathcal{D}_{t+1})\|^2 + \frac{L\eta^2}{2}\mathbb{E}\left[\|\nabla f(\theta_t, \zeta_{t+1})\|^2\right]$$

$$= -\eta P_{\min}^\pi\mathbb{E}\|\nabla f(\theta_t, \mathcal{D}_{t+1})\|^2 + \frac{L\eta^2}{2}\mathbb{E}\left[\|\nabla f(\theta_t, \zeta_{t+1}) - \nabla f(\theta_t, \mathcal{D}_{t+1}) + \nabla f(\theta_t, \mathcal{D}_{t+1})\|^2\right]$$

$$\leq -\eta P_{\min}^\pi\mathbb{E}\|\nabla f(\theta_t, \mathcal{D}_{t+1})\|^2 + \frac{L\eta^2}{2}\left(\mathbb{E}\|\nabla f(\theta_t, \zeta_{t+1}) - \nabla f(\theta_t, \mathcal{D}_{t+1})\|^2 + \mathbb{E}\|\nabla f(\theta_t, \mathcal{D}_{t+1})\|^2\right)$$

$$\overset{(g)}{\leq} -\eta P_{\min}^\pi\mathbb{E}\|\nabla f(\theta_t, \mathcal{D}_{t+1})\|^2 + \frac{L\eta^2}{2}\sigma^2 + \frac{L\eta^2}{2}\mathbb{E}\|\nabla f(\theta_t, \mathcal{D}_{t+1})\|^2$$

$$= \left(\frac{L\eta^2}{2} - \eta P_{\min}^\pi\right)\mathbb{E}\|\nabla f(\theta_t, \mathcal{D}_{t+1})\|^2 + \frac{L\eta^2}{2}\sigma^2, \tag{17}$$

where (g) follows from **Assumption 2** on the bounded variance of stochastic gradients.

By adding $\mathbb{E}\left[f(\theta_t, \mathcal{D}_{t+1}) - f(\theta_t, \mathcal{D}_t)\right] = \mathbb{E}\left[\Delta f_\delta(t)\right]$ on both sides of Equation (17), the expression becomes:

$$\mathbb{E}[f(\theta_{t+1}, \mathcal{D}_{t+1}) - f(\theta_t, \mathcal{D}_t)] \leq \left(\frac{L\eta^2}{2} - \eta P_{\min}^\pi\right) \mathbb{E}\left\|\nabla f(\theta_t, \mathcal{D}_{t+1})\right\|^2 + \frac{L\eta^2}{2}\sigma^2 + \mathbb{E}\left[\Delta f_\delta(t)\right]. \tag{18}$$

Now note that all expectations are of values calculated for the entire dataset at every time, so it is true that: $\mathbb{E}[f(\theta_{t+1}, \mathcal{D}_{t+1})] = f(\theta_{t+1}, \mathcal{D}_{t+1})$, $\mathbb{E}[f(\theta_t, \mathcal{D}_t)] = f(\theta_t, \mathcal{D}_t)$, $\mathbb{E}\left[\Delta f_\delta(t)\right] = \Delta f_\delta(t)$, and $\mathbb{E}\left\|\nabla f(\theta_t, \mathcal{D}_{t+1})\right\|^2 = \left\|\nabla f(\theta_t, \mathcal{D}_{t+1})\right\|^2$.

Utilizing this fact and rearranging the terms, the bound on the gradient becomes:

$$\left\|\nabla f(\theta_t, \mathcal{D}_{t+1})\right\|^2 \leq \frac{1}{\left(P_{\min}^\pi \eta - \frac{L\eta^2}{2}\right)} \left(f(\theta_t, \mathcal{D}_t) - f(\theta_{t+1}, \mathcal{D}_{t+1})\right)$$
$$+ \frac{\Delta f_\delta(t)}{\left(P_{\min}^\pi \eta - \frac{L\eta^2}{2}\right)} + \frac{L\eta}{2P_{\min}^\pi - L\eta}\sigma^2. \tag{19}$$

Finally, using $\mu := \left(P_{\min}^\pi \eta - \frac{L\eta^2}{2}\right)$, by telescoping over and taking a time average, the expression becomes:

$$\frac{1}{T}\sum_{t=0}^{T-1}\left\|\nabla f(\theta_t, \mathcal{D}_{t+1})\right\|^2 \leq \frac{1}{\mu T}\left(f(\theta_0, \mathcal{D}_0) - f(\theta_T, \mathcal{D}_T) + \sum_{t=0}^{T-1}\Delta f_\delta(t)\right)$$
$$+ \frac{L\eta}{2P_{\min}^\pi - L\eta}\sigma^2. \tag{20}$$

proving **Theorem 5.1**. $\qquad\square$

### B.2 Proof of Corollary 5.2

We can relate the drift-induced loss term with the dynamic bound on the concept drift as follows:

$$\begin{aligned}
\Delta f_\delta(t) &= f(\theta_t, \mathcal{D}_{t+1}) - f(\theta_t, \mathcal{D}_t) \\
&\overset{(a)}{=} \mathbb{E}_{\omega \sim \mathcal{D}_{t+1}}\left[f(\theta_t, \omega)\right] - \mathbb{E}_{\omega \sim \mathcal{D}_t}\left[f(\theta_t, \omega)\right] \\
&\overset{(b)}{=} \sum_{\omega_i} P_{\mathcal{D}_{t+1}}(\omega_i)f(\theta_t, \omega_i) - \sum_{\omega_i} P_{\mathcal{D}_t}(\omega_i)f(\theta_t, \omega_i) \\
&\overset{(c)}{\leq} \sum_{\omega_i}\left|P_{\mathcal{D}_{t+1}}(\omega_i) - P_{\mathcal{D}_t}(\omega_i)\right| B \\
&\overset{(d)}{=} \left\|\mathbf{P}_{\mathcal{D}_t} - \mathbf{P}_{\mathcal{D}_{t+1}}\right\|_1 B \\
&\overset{(e)}{\leq} B\sqrt{2\mathrm{D}_{\mathrm{KL}}\left(\mathcal{D}_t \| \mathcal{D}_{t+1}\right)} \\
&\overset{(f)}{\leq} B\sqrt{2\delta(t)}, 
\end{aligned} \tag{21}$$

where:

(a) follows from **Assumption 2** on the unbiased estimate of the true gradient.

(b) uses the definition of expectation over a discrete probability space. The sample space $\{\omega_i\}$ is the set of all possible batches that can be formed from the union of all datasets $\bigcup_{t=0}^{T-1}\mathcal{D}_t$. Since this union is finite, the set $\{\omega_i\}$ is also finite. $P_{\mathcal{D}_t}(\omega_i)$ is the probability of sampling batch $\omega_i$ from the dataset $\mathcal{D}_t$, where $P_{\mathcal{D}_t}(\omega_i) = 0$ if $\omega_i$ cannot be formed from $\mathcal{D}_t$. This probability distribution $P_{\mathcal{D}_t}$ varies over time due to concept drift.

(c) is true by **Assumption 4** on the boundedness of the loss, and the fact that $\forall x \in \mathbb{R}, x \leq |x|$.

(d) follows from the definition of the L1 loss, where $\mathbf{P}_{\mathcal{D}_t}$ is the probability distribution vector, containing probabilities of sampling batches $\omega_i$ $\forall i$ from the dataset $\mathcal{D}_t$.

(e) is true by the Pinsker's Inequality.

(f) follows from **Assumption 3** on the boundedness of the concept drift at different times $t$.

Now, by **Assumption 4** on the boundedness of the loss:

$$f(\theta_0, \mathcal{D}_0) - f(\theta_T, \mathcal{D}_T) \leq f(\theta_0, \mathcal{D}_0) \leq B. \tag{22}$$

By applying the results of Equations 21 and 22 to **Theorem 5.1** and utilizing the definition of $\mu$, the bound becomes:

$$\frac{1}{T} \sum_{t=0}^{T-1} \|\nabla f(\theta_t, \mathcal{D}_{t+1})\|^2 \leq \frac{B}{T \left( \eta P_{\min}^\pi - \frac{L\eta^2}{2} \right)} \left( 1 + \sqrt{2} \sum_{t=0}^{T-1} \sqrt{\delta(t)} \right) + \frac{L\eta}{2 P_{\min}^\pi - L\eta} \sigma^2. \tag{23}$$

This completes the proof of **Corollary 5.2**. $\qquad\qquad\qquad\qquad\qquad\qquad\qquad\qquad\qquad\qquad\qquad\square$

## C  Lyapunov Analysis

### C.1  Lyapunov Drift-Plus-Penalty Framework

The convergence result depends on two key components bounding the loss gradients over time: the telescoped loss $f(\theta_0, \mathcal{D}_0) - f(\theta_T, \mathcal{D}_T)$ and the drift-induced loss $\Delta f_\delta(t)$. By undoing the telescoping step, and utilizing the inference losses (which are an unbiased estimate of the true loses that use the entire dataset $\mathcal{D}_t$ for computation), we can define the effective policy, as a policy that solves the following problem:

$$\min \ \frac{1}{T} \sum_{t=0}^{T-1} \underbrace{f(\theta_t, \xi_t) - f(\theta_{t+1}, \xi_{t+1})}_{\text{model evolution loss under drift}} + (1 - \pi(t)) \underbrace{\Delta f_{\delta,\xi}(t)}_{\text{drift loss}}.$$

$$\text{subject to } \frac{1}{T} \sum \lambda(t)\pi(t) \leq \bar{\lambda}. \tag{24}$$

Note that the sum utilized in this optimization problem differs from the one in Theorem 5.1, as we use the inference losses, and we introduced a $(1 - \pi(t))$ multiplier of $\Delta f_{\delta,\xi}(t)$. The inference losses are used as that's what the agent can access at different time steps $t$. The $(1 - \pi(t))$ was introduced for two reasons:

- It emphasizes the penalty associated with no update under strong drift, and rewards the updates better.
- If the model is updated, $\Delta f_\delta(t)$ effectively corresponds to a theoretical loss increase that would have happened had the model parameters not been updated. By introducing $(1 - \pi(t))$, we exclude this theoretical increase from optimization, placing emphasis only on observed loss values.

While this modification results in a different policy compared to one derived from the unmodified penalty, $\frac{1}{T} \sum_{t=0}^{T-1} f(\theta_t, \mathcal{D}_t) - f(\theta_{t+1}, \mathcal{D}_{t+1}) + \Delta f_\delta(t)$, it is important to recognize that applying the drift-plus-penalty framework to either problem formulation would not solve the long-term minimization directly. The Lyapunov method, by design, converts a long-term stochastic optimization problem into a series of deterministic, per-time-slot minimizations. This greedy, "one-shot" decision is inherently an approximation of the true optimal policy that would minimize the sum over the entire horizon $\mathcal{T}$.

Given that the per-slot minimization is already a well-motivated heuristic, we are justified in designing the per-slot penalty function to be a more effective proxy for our control goals. Our modification, which explicitly isolates the drift-loss penalty to the $\pi(t) = 0$ ("no-update") action, simply provides a more targeted and intuitive objective for the greedy controller to optimize at each step.

With the modified penalty, the optimization problem we aim to solve is:

$$\min \frac{1}{T} \sum_{t=0}^{T-1} f(\theta_t, \xi_t) - f(\underbrace{\theta_t - \pi(t+1)\eta\nabla f(\theta_t, \zeta_{t+1})}_{\theta_{t+1}}, \xi_{t+1}) + (1 - \pi(t))\Delta f_{\delta,\xi}(t),$$

$$\text{subject to } \frac{1}{T} \sum \lambda(t)\pi(t) \leq \bar{\lambda}. \tag{25}$$

Finding the optimal solution that minimizes entire sum is hard due to the evolving nature of the environment. A truly optimal policy would require knowledge of the future, as given full information about the evolution of the dataset over the entire time horizon, the optimal points to retrain will depend on all $t$. This requirement makes the problem intractable to solve directly. As such, we utilize the Lyapunov analysis framework to derive an online policy that does not require the knowledge of the entire environment evolution. We begin by defining the virtual queue as:

$$Q(t+1) = \max\left\{0, Q(t) + \lambda(t)\pi(t) - \bar{\lambda}\right\}, \tag{26}$$

from which it follows that the Lyapunov drift is:

$$\Delta Q(t) = \frac{1}{2}\left[Q(t+1)^2 - Q(t)^2\right] \leq \left[Q(t)\left(\lambda(t)\pi(t) - \bar{\lambda}\right) + \frac{(\lambda(t)\pi(t) - \bar{\lambda})^2}{2}\right]. \tag{27}$$

The drift-plus-penalty term is bounded as:

$$\mathbb{E}\left[\Delta Q(t) \mid Q(t)\right] + V \times \Psi(t) \leq \mathbb{E}\left[Q(t)\left(\lambda(t)\pi(t) - \bar{\lambda}\right) + \frac{(\lambda(t)\pi(t) - \bar{\lambda})^2}{2}\bigg| Q(t)\right]$$

$$+ V\left(f(\theta_t, \xi_t) - f(\theta_t - \pi(t+1)\eta\nabla f(\theta_t, \zeta_{t+1}), \xi_{t+1}) + (1 - \pi(t))\Delta f_{\delta,\xi}(t)\right)$$

$$= \mathbb{E}\left[Q(t)\left(\lambda(t)\pi(t) - \bar{\lambda}\right) + \frac{(\lambda(t)\pi(t) - \bar{\lambda})^2}{2}\bigg| Q(t)\right] + Vf(\theta_t, \xi_t)$$

$$- Vf(\theta_t - \pi(t+1)\eta\nabla f(\theta_t, \zeta_{t+1}), \xi_{t+1}) + V(1 - \pi(t))f(\theta_t, \xi_{t+1}) - V(1 - \pi(t))f(\theta_t, \xi_t) \tag{28}$$

where $\Psi(t)$ is the penalty term at any time $t$, which corresponds to the inference loss increase due to the evolution of the environment $f(\theta_t, \xi_t) - f(\theta_t - \pi(t+1)\eta\nabla f(\theta_t, \zeta_{t+1}), \xi_{t+1})$ combined with the drift-induced inference loss increase when not updating the model $(1 - \pi(t))\Delta f_{\delta,\xi}(t)$.

Next, as is standard in Lyapunov analysis, we solve the problem on a per-time-slot basis. The problem to solve at each time $t$ becomes (note that the expectation is dropped, because for a given value of $\pi(t)$, all values in it are deterministic):

$$\min_{\pi \in \{0,1\}} Q(t)\left(\lambda(t)\pi(t) - \bar{\lambda}\right) + \frac{(\lambda(t)\pi(t) - \bar{\lambda})^2}{2} + Vf(\theta_t, \xi_t)$$

$$- Vf(\theta_t - \pi(t+1)\eta\nabla f(\theta_t, \zeta_{t+1}), \xi_{t+1}) + (1 - \pi(t))V\Delta f_{\delta,\xi}(t).$$

Then, if $\pi(t) = 0$, we have $\theta_t = \theta_{t-1}$ and the immediate bound becomes:

$$-\bar{\lambda}Q(t) + \frac{\bar{\lambda}^2}{2} + Vf(\theta_{t-1}, \xi_{t+1}) - Vf(\theta_{t-1} - \pi(t+1)\eta\nabla f(\theta_{t-1}, \zeta_{t+1}), \xi_{t+1}).$$

Otherwise, if $\pi(t) = 1$, we have $\theta_t = \theta_{t-1} - \eta\nabla_\theta f(\theta_{t-1}, \zeta_t)$ and the immediate bound becomes:

$$\lambda(t)Q(t) - \bar{\lambda}Q(t) + \frac{(\lambda(t) - \bar{\lambda})^2}{2} + Vf(\theta_{t-1} - \eta\nabla f(\theta_{t-1}, \zeta_t), \xi_t)$$

$$- Vf(\theta_{t-1} - \eta\nabla f(\theta_{t-1}, \zeta_t) - \pi(t+1)\eta\nabla f(\theta_{t-1} - \eta\nabla f(\theta_{t-1}, \zeta_t), \zeta_{t+1}), \xi_{t+1}).$$

By comparing the two resulting costs, the policy should update if the cost associated with not updating is greater than or equal to the cost of updating:

$$-\bar{\lambda}Q(t) + \frac{\bar{\lambda}^2}{2} + Vf(\theta_{t-1}, \xi_{t+1}) - Vf(\theta_{t-1} - \pi(t+1)\eta\nabla f(\theta_{t-1}, \zeta_{t+1}), \xi_{t+1})$$

$$\geq \lambda(t)Q(t) - \bar{\lambda}Q(t) + \frac{(\lambda(t) - \bar{\lambda})^2}{2} + Vf(\theta_{t-1} - \eta\nabla f(\theta_{t-1}, \zeta_t), \xi_t)$$

$$- Vf(\theta_{t-1} - \eta\nabla f(\theta_{t-1}, \zeta_t) - \pi(t+1)\eta\nabla f(\theta_{t-1} - \eta\nabla f(\theta_{t-1}, \zeta_t), \zeta_{t+1}), \xi_{t+1})$$

By rearranging, we get

$$Vf(\theta_{t-1}, \xi_{t+1}) - Vf(\theta_{t-1} - \pi(t+1)\eta\nabla f(\theta_{t-1}, \zeta_{t+1}), \xi_{t+1})$$
$$+ Vf(\theta_{t-1} - \eta\nabla f(\theta_{t-1}, \zeta_t) - \pi(t+1)\eta\nabla f(\theta_{t-1} - \eta\nabla f(\theta_{t-1}, \zeta_t), \zeta_{t+1}), \xi_{t+1})$$
$$- Vf(\theta_{t-1} - \pi(t+1)\eta\nabla f(\theta_{t-1}, \zeta_{t+1}), \xi_{t+1})$$
$$\geq \lambda(t)Q(t) + \frac{1}{2}\left[\lambda(t)^2 - 2\bar{\lambda}\lambda(t)\right].$$

For the simplicity of notations, we define

$$\mathcal{J}_\pi(t) := f(\theta_{t-1} - \eta\nabla f(\theta_{t-1}, \zeta_t) - \pi(t+1)\eta\nabla f(\theta_{t-1} - \eta\nabla f(\theta_{t-1}, \zeta_t), \zeta_{t+1}), \xi_{t+1})$$
$$- f(\theta_{t-1} - \pi(t+1)\eta\nabla f(\theta_{t-1}, \zeta_{t+1}), \xi_{t+1}).$$

Further, by rearranging, we arrive at the update rule proposed in (9):

$$V\left(f(\theta_{t-1}, \xi_{t+1}) - f(\theta_{t-1} - \eta\nabla f(\theta_{t-1}, \zeta_t), \xi_t)\right) + V\mathcal{J}_\pi(t) \geq \lambda(t)Q(t) + \frac{1}{2}\left[\lambda(t)^2 - 2\bar{\lambda}\lambda(t)\right].$$

This completes the analysis.

Note that $\mathcal{J}_\pi(t)$ corresponds to the future impact on the loss of the current decision, $\pi(t)$. The choice of $\pi(t)$ directly affects the virtual queue length $Q(t+1)$, altering the state observed by the policy at time $t+1$ and influencing the next decision $\pi(t+1)$. This then changes the expected value of the future model, $\mathbb{E}[\theta_{t+1}]$. As a result, $\mathcal{J}_\pi(t)$ couples the immediate, greedy decision at time $t$ with its expected consequence on the loss at time $t+1$. It is intractable to compute in a practical online system, as it depends on future values of environmental variables. The same is true for the $f(\theta_{t-1}, \xi_{t+1}) - f(\theta_{t-1} - \eta\nabla f(\theta_{t-1}, \zeta_t), \xi_t)$ term (where while computing the gradient at every time step $t$ is feasible, it would violate the resource constraint). As a result, while both terms are crucial for the theoretical derivation, in our practical implementation of this policy we use an approximation for

$$\left(f(\theta_{t-1}, \xi_{t+1}) - f(\theta_{t-1} - \eta\nabla f(\theta_{t-1}, \zeta_t), \xi_t)\right) + \mathcal{J}_\pi(t).$$

## C.2 Stability Analysis: Proof of Theorem 5.3

Let $\pi^*$ denote our proposed policy. At each time slot $t$, the policy was chosen to minimize

$$Q(t)\left(\lambda(t)\pi(t) - \bar{\lambda}\right) + \frac{(\lambda(t)\pi(t) - \bar{\lambda})^2}{2} + Vf(\theta_t, \xi_t) - Vf(\theta_{t+1}, \xi_{t+1}) + (1 - \pi(t))V\Delta f_{\delta,\xi}(t).$$

Next, consider a policy $\pi_{\text{Uniform}}$ such that $\pi_{\text{Uniform}}(t) = 1$ with probability $\min\left(\frac{\bar{\lambda}}{\lambda(t)}, 1\right)$

Then, for all $t$, the following holds

$$\mathbb{E}\left[Q(t)\left(\lambda(t)\pi^*(t) - \bar{\lambda}\right) + \frac{1}{2}\left(\lambda(t)\pi^*(t) - \bar{\lambda}\right)^2 + Vf(\theta_t, \xi_t) - Vf(\theta_{t+1}, \xi_{t+1})\right.$$
$$\left. + (1 - \pi^*(t))V\Delta f_{\delta,\xi}(t)\right]$$

$$\leq \mathbb{E}\left[Q(t)\left(\lambda(t)\pi_{\text{Uniform}}(t) - \bar{\lambda}\right) + \frac{1}{2}\left(\lambda(t)\pi_{\text{Uniform}}(t) - \bar{\lambda}\right)^2 + Vf(\theta_t, \xi_t) - Vf(\theta_{t+1}, \xi_{t+1})\right.$$

$$\left. + (1 - \pi_{\text{Uniform}}(t))V\Delta f_\delta(t)\right]. \tag{29}$$

Next note that:

$$\mathbb{E}\left[Q(t)\left(\lambda(t)\pi_{\text{Uniform}}(t) - \bar{\lambda}\right) \mid Q(t)\right] = \mathbb{E}\left[Q(t)\lambda(t)\pi_{\text{Uniform}}(t) \mid Q(t)\right] - \mathbb{E}\left[Q(t)\bar{\lambda} \mid Q(t)\right]$$
$$= Q(t)\lambda(t)\mathbb{E}[\pi_{\text{Uniform}}(t) \mid Q(t)] - \bar{\lambda}Q(t)$$
$$= Q(t)\lambda(t)\frac{\bar{\lambda}}{\lambda(t)} - \bar{\lambda}Q(t)$$

$$= 0. \tag{30}$$

Then, using the same principle as above, as well as the fact that $\mathbb{E}[\pi(t)] = \mathbb{E}[\pi(t)^2]$, it follows that:

$$\mathbb{E}\left[\frac{1}{2}\left(\lambda(t)\pi_{\text{Uniform}}(t) - \bar{\lambda}\right)^2 \,\Big|\, Q(t)\right] = \frac{1}{2}\bar{\lambda}\left(\lambda(t) - \bar{\lambda}\right).$$

Using the above, Assumptions 3-4, and introducing $\mathbb{E}_{\pi^*}[\cdot]$ and $\mathbb{E}_{\pi_{\text{Uniform}}}[\cdot]$ as the expectation over the drift-plus-penalty given $Q(t)$ for policies $\pi^*$ and $\pi_{\text{Uniform}}$, respectively, the inequality becomes:

$$
\begin{aligned}
\mathbb{E}_{\pi^*} & \left[Q(t)\left(\lambda(t)\pi^*(t) - \bar{\lambda}\right) + \frac{1}{2}\left(\lambda(t)\pi^*(t) - \bar{\lambda}\right)^2\right] \\
\le & \frac{1}{2}\bar{\lambda}\left(\lambda(t) - \bar{\lambda}\right) - V\mathbb{E}_{\pi^*}\left[f(\theta_t, \xi_t)\right] + V\mathbb{E}_{\pi^*}\left[f(\theta_{t+1}, \xi_{t+1})\right] \\
& + V\mathbb{E}_{\pi_{\text{Uniform}}}\left[f(\theta_t, \xi_t)\right] - V\mathbb{E}_{\pi_{\text{Uniform}}}\left[f(\theta_{t+1}, \xi_{t+1})\right] \\
& + V\mathbb{E}_{\pi_{\text{Uniform}}}\left[\Delta f_{\delta,\xi}(t)\right] - V\mathbb{E}_{\pi_{\text{Uniform}}}\left[\min\left(\frac{\bar{\lambda}}{\lambda(t)}, 1\right)\Delta f_{\delta,\xi}(t)\right] \\
& - V\mathbb{E}_{\pi^*}\left[\Delta f_{\delta,\xi}(t)\right] + V\mathbb{E}_{\pi^*}\left[\pi^*(t)\Delta f_{\delta,\xi}(t)\right] \\
\le & \frac{1}{2}\bar{\lambda}\left(\lambda(t) - \bar{\lambda}\right) + 2VB + VB\sqrt{2\delta(t)} + 3VB \\
= & \frac{1}{2}\bar{\lambda}\left(\lambda(t) - \bar{\lambda}\right) + 5VB + VB\sqrt{2\delta(t)}. \tag{31}
\end{aligned}
$$

By taking the total expectation, and by definition of the Lyapunov drift:

$$\mathbb{E}[\Delta Q(t)] \le \frac{1}{2}\bar{\lambda}\left(\lambda(t) - \bar{\lambda}\right) + 5VB + VB\sqrt{2\delta(t)}.$$

Next, we have

$$\frac{1}{2}Q(T)^2 - \frac{1}{2}Q(0)^2 = \sum_{t=0}^{T-1}\Delta Q(t)$$

$$\Rightarrow \frac{1}{2}\mathbb{E}\left[Q(T)^2\right] - \frac{1}{2}\mathbb{E}\left[Q(0)^2\right]$$

$$= \sum_{t=0}^{T-1}\mathbb{E}\left[\Delta Q(t)\right] \le \frac{1}{2}\bar{\lambda}\left(\sum_{t=0}^{T-1}\lambda(t) - T\bar{\lambda}\right) + \left(5TVB + VB\sum_{t=0}^{T-1}\sqrt{2\delta(t)}\right)$$

$$\Rightarrow \mathbb{E}\left[Q(T)\right] \le \sqrt{\mathbb{E}\left[Q(T)^2\right]} \le \sqrt{\mathbb{E}[Q(0)] + \bar{\lambda}\left(\sum_{t=0}^{T-1}\lambda(t) - T\bar{\lambda}\right) + 2VB\left(5T + \sum_{t=0}^{T-1}\sqrt{2\delta(t)}\right)},$$

and assuming $Q(0) = 0$, we arrive at an intermediate bound on the expected queue length at time $T$:

$$\mathbb{E}\left[Q(T)\right] \le \sqrt{\bar{\lambda}\left(\sum_{t=0}^{T-1}\lambda(t) - T\bar{\lambda}\right) + 2VB\left(5T + \sum_{t=0}^{T-1}\sqrt{2\delta(t)}\right)}. \tag{32}$$

Then, $\lambda(t)\pi(t) - \bar{\lambda} \le Q(t+1) - Q(t)$ implies

$$\frac{1}{T}\sum_{t=0}^{T-1}\lambda(t)\mathbb{E}\left[\pi(t)\right] - \bar{\lambda} \le \frac{1}{T}\mathbb{E}[Q(T)] \le \sqrt{\frac{\bar{\lambda}}{T^2}\sum_{t=0}^{T-1}\lambda(t) - \frac{\bar{\lambda}^2}{T} + 2VB\left(\frac{5}{T} + \frac{1}{T^2}\sum_{t=0}^{T-1}\sqrt{2\delta(t)}\right)}.$$

So, the final inequality becomes:

$$\frac{1}{T}\sum_{t=0}^{T-1}\lambda(t)\mathbb{E}\left[\pi(t)\right] - \bar{\lambda} \le \sqrt{\frac{\bar{\lambda}}{T^2}\sum_{t=0}^{T-1}\lambda(t) - \frac{\bar{\lambda}^2}{T} + 2VB\left(\frac{5}{T} + \frac{1}{T^2}\sum_{t=0}^{T-1}\sqrt{2\delta(t)}\right)}. \tag{33}$$

This completes the proof Theorem 5.3. $\qquad\square$

# D   Computational Resource Evaluation

In this section, we provide a detailed evaluation of the computational resources associated with the operation of RCCDA. We first discuss the abstract nature of "resources" assumed in our framework to demonstrate its generalization. We then conduct an overhead analysis to quantify the additional resource consumption of RCCDA itself compared to the evaluated baseline policies and discuss its efficiency relative to traditional drift detection methods.

## D.1   Resource Abstraction and Generalizability

Throughout this paper, the concept a "resource" is treated abstractly. This is a deliberate design choice to ensure that our framework is broadly applicable to a wide range of real-world, resource-constrained scenarios. A resource can represent any quantifiable commodity that is consumed during a model update and is subject to a limited budget. This could be computational cycles, FLOPs, power consumption, data usage, or wall-clock time. At any time step $t$, the agent can perform a model update, which incurs a general, time-varying cost, $\lambda(t)$, associated with the considered resource type. The agent's operational constraint is a total resource budget over the entire time horizon, which is enforced via the time-average resource constraint, $\bar{\lambda}$, used to bound resource usage in Equation (5). This is equivalent to ensuring the total amount of consumed resources, $\sum_{t=0}^{T} \lambda(t)\pi(t)$, does not exceed the total available resource budget, $T\bar{\lambda}$.

To demonstrate that this approach is generalizable across different resource types, we conducted a new empirical analysis to quantify the average resource usage per update for several distinct metrics. In our setup, we utilized the same experimental settings as in our main experiments on the PACS dataset, details available in Appendix E, averaging the results over 250 updates. We measured the resource usage for a single model update across the following metrics: computation (GFLOPs), data processed (GB), memory-time usage (GB-s), energy consumption (Joules), and wall-clock time (s).

Table 2 reports the empirically measured average cost per update $\lambda(t)$, and the corresponding total available resource budget $T\bar{\lambda}$ for various metrics.

Table 2: Empirical analysis of resource consumption for different resource types for the main experiment on the PACS dataset.

| Resource Metric Type (Unit) | Constant Average Update Cost | Total Available Budget |
|---|---|---|
| Computation (GFLOPs) | $11438.0 \pm 0.0$ | 285,951.0 |
| Data Processed (GB) | $0.11719 \pm 0.0$ | 2.92975 |
| Memory-Time (GB-s) | $20.11 \pm 0.3$ | 502.75 |
| Energy (Joules) | $626.67 \pm 24.35$ | 15,666.75 |
| Wall-Clock Time (s) | $2.174 \pm 0.037$ | 54.35 |

Note that RCCDA performs identically for all the resource metrics in Table 2, as the experiments were configured to use the same ratio $\frac{\bar{\lambda}}{\lambda(t)}$ (with $\lambda(t)$ set to a constant value $\lambda$), effectively defining a target update frequency constraint. Since the decision logic in RCCDA is driven by this ratio and the observed model loss, its behavior remains consistent regardless of how the "resource" is defined. This result therefore confirms that our framework is generalizable to different resource metrics, demonstrating that RCCDA is a versatile solution for resource-constrained adaptation.

## D.2   Overhead Analysis of the RCCDA Policy

A critical aspect of designing a model update policy for resource-constrained environments is ensuring the decision-making process itself does not introduce significant computational overhead. Many existing approaches to mitigating concept drift rely on explicit drift detection mechanisms, which makes them computationally intensive and ill-suited for real-time decision making.

To demonstrate the necessity for a lightweight policy that does not rely on explicit drift detection, we conducted an empirical analysis comparing RCCDA against two established drift detection methods: the Adaptive Windowing (ADWIN) [27] algorithm and the Kolmogorov-Smirnov (KS) [61] test. Traditional drift detection methods like these typically require constant monitoring of

the statistical properties of the entire incoming data stream (KS-Test) or sequentially processing error rates (ADWIN). In contrast, RCCDA relies solely on the model's inference loss, which is a lightweight scalar value, often computed or estimated as part of the model's primary task evaluation. The reliance on a pre-existing simple signal makes our approach inherently efficient.

To quantify this efficiency, we measured the decision-making wall-clock time taken by RCCDA, and compared it to a wall-clock time taken by ADWIN and KS-Test to finish performing drift detection operations, which is a required step if using these algorithms for decision making. We assume all necessary inputs (such as the inference loss for RCCDA, data windows for the others) are already available through running the initial model evaluation. The experiment was conducted using the PACS dataset with the setup detailed in Appendix E, averaging results over 4 domains and 3 different seeds.

The results in Table 3 demonstrate the computational efficiency of our approach. RCCDA's decision time is negligible, measuring in the sub-microsecond range. In contrast, ADWIN is over three orders of magnitude slower. While small in isolation, especially compared to the model inference time, this latency would accumulate over a long operational horizon, increasing both response delay and cumulative resource consumption. The KS-Test proves computationally prohibitive, exceeding RCCDA's decision time by over eight orders of magnitude due to the requirement of performing statistical tests on the entire data window. These vast time disparities underscore a conclusion that policies reliant on explicit drift detection introduce a temporal overhead that is unsustainable in many resource-constrained scenarios, validating our lightweight, loss-driven approach.

Table 3: Comparison of additional decision-making time for RCCDA versus established drift detection methods.

| Algorithm | Additional Decision Making Time (ms) | Relative Slowdown (vs. RCCDA) |
|---|---|---|
| RCCDA | $0.000744 \pm 0.000142$ | 1 |
| ADWIN | $1.845618 \pm 0.469732$ | $\sim 2480$ |
| KS-Test | $125996.5 \pm 9605.7$ | $\sim 169,350,134$ |

While this comparison demonstrates that the traditional drift detection methods add a significant computational overhead from monitoring data streams or performance metrics, ADWIN and the KS-Test are not policy algorithms by design, and therefore are not designed to operate under a strict resource budget. As such, a more direct and informative benchmark for policy-specific overhead comes from comparing RCCDA against the baseline methods from the main experiments, which are explicitly designed for operation under resource constraints. These policies, such as Uniform or Periodic updates, are inherently lightweight and serve as an excellent benchmark for verifying that RCCDA does not introduce a prohibitive computational and resource cost relative to simple heuristics. We therefore evaluated the total memory footprint, decision time, and theoretical complexity for each policy to provide a comprehensive overhead analysis. The experiment was conducted on the PACS dataset, with a window size of $w = 40$ used by Budget-Increase and Budget-Threshold policies. The results are summarized in Table 4. As shown, RCCDA's resource consumption is comparable to the baseline policies on all evaluated metrics.

Table 4: Overhead comparison of RCCDA against baseline policies. All policies exhibit minimal overhead, with RCCDA remaining competitive while providing robust adaptation.

| Policy | Total Memory (bytes) | Wall-Clock Time ($\mu s$) | Complexity |
|---|---|---|---|
| RCCDA | 276 | 1.94 | $\mathcal{O}(1)$ |
| Uniform Random | 72 | 1.79 | $\mathcal{O}(1)$ |
| Periodic | 108 | 0.99 | $\mathcal{O}(1)$ |
| Budget-Increase | $212 + 32w$ | 1.74 | $\mathcal{O}(w)$ |
| Budget-Threshold | $212 + 32w$ | 1.68 | $\mathcal{O}(w)$ ($\mathcal{O}(1)$ optimized) |

This two-fold resource analysis demonstrates that RCCDA is both vastly superior to traditional drift detectors and is competitively efficient against the lightweight baselines, confirming that our policy successfully integrates an intelligent, adaptive decision-making mechanism without incurring a prohibitive operational cost.

# E   Detailed Experimental Setup

In this section, we describe the experimental setup in detail. Our main experiment consists of two major phases. In the *pretraining phase* (described in E.3), we train the models on selected source domains from the datasets until convergence. In the subsequent *evaluation phase* (described in E.4), the agent deploys the pretrained model in an environment that simulates concept drift by introducing new domains, and uses a retraining policy to mitigate that drift.

## E.1   Datasets

To simulate concept drift in our experiments, we used 4 common domain generalization (DG) datasets:

- **PACS** [62]: PACS consists of images from 4 domains: Photo, Art Painting, Cartoon, and Sketch, each with the 7 common categories: dog, elephant, giraffe, guitar, horse, house, and person. The default size of the images is $224 \times 224$. We transform all images into $128 \times 128$.

- **DigitsDG** [55]: DigitsDG encompasses images of handwritten digits 0 to 9, sourced from 4 different domains: SVHN, SYN, MNIST, and MNIST-M. The image sizes are $32 \times 32$.

- **OfficeHome** [56]: OfficeHome consists of images from four domains: Art, Clipart, Product, and Real-World, each featuring 65 object categories commonly encountered in office and home environments, including chairs, desks, and household items. The image sizes are $224 \times 224$.

- **MEMD-ABSA** [57]: MEMD-ABSA is a multi-domain textual dataset for Aspect-Based Sentiment Analysis (ABSA). It consists of approximately 20,000 sentences from 5 distinct domains: Books, Clothing, Hotel, Laptop and Restaurant. Each sentence annotated with one or more sentiment quadruples: aspect, category, opinion, and sentiment. The sentiment is then classified as positive, negative, or neutral.

## E.2   Models

Below, we specify the models employed for each dataset:

**PACS**. For the PACS dataset, we use PACSCNN, a custom convolutional neural network (CNN) with skip connections. The model has 11,177,223 parameters, and its architecture is detailed below:

| Layer | Layer Description |
|-------|-------------------|
| 1 | Convolutional layer: 3 input channels, 64 output channels, kernel size 3, stride 1, padding 1 |
| 2 | Batch normalization layer |
| 3 | ReLU activation |
| 4 | Residual block: 64 input channels, 64 output channels, stride 1 |
| 5 | Residual block: 64 input channels, 64 output channels, stride 1 |
| 6 | Residual block: 64 input channels, 128 output channels, stride 2 |
| 7 | Residual block: 128 input channels, 128 output channels, stride 1 |
| 8 | Residual block: 128 input channels, 256 output channels, stride 2 |
| 9 | Residual block: 256 input channels, 256 output channels, stride 1 |
| 10 | Residual block: 256 input channels, 512 output channels, stride 2 |
| 11 | Residual block: 512 input channels, 512 output channels, stride 1 |
| 12 | Adaptive average pooling layer: reduces spatial dimensions to $1 \times 1$ |
| 13 | Flatten operation |
| 14 | Dropout layer: $p = 0.5$ |
| 15 | Linear layer: 512 input features to 7 output classes |

Each residual block has the following structure:

| Layer | Layer Description |
|---|---|
| 1 | Convolutional layer: input channels to output channels, stride passed as argument, kernel size 3, padding 1 |
| 2 | Batch normalization layer |
| 3 | ReLU activation |
| 4 | Convolutional layer: output channels to output channels, kernel size 3, stride 1, padding 1 |
| 5 | Batch normalization layer |
| 6 | Skip connection: identity mapping if input channels equal output channels and stride=1; otherwise, a 1x1 convolutional projection with the given stride followed by batch normalization |
| 7 | ReLU activation |

**DigitsDG**. For the DigitsDG dataset, we use DigitsDGCNN, a CNN model with 1,907,146 parameters, and the following architecture:

| Layer | Layer Description |
|---|---|
| 1 | Convolutional layer: 3 input channels, 64 output channels, kernel size 3, stride 1, padding 1 |
| 2 | Batch normalization layer |
| 3 | ReLU activation |
| 4 | Convolutional layer: 64 input channels, 64 output channels, kernel size 1, stride 2, padding 0 |
| 5 | Convolutional layer: 64 input channels, 128 output channels, kernel size 3, stride 1, padding 1 |
| 6 | Batch normalization layer |
| 7 | ReLU activation |
| 8 | Convolutional layer: 128 input channels, 128 output channels, kernel size 1, stride 2, padding 0 |
| 9 | Convolutional layer: 128 input channels, 256 output channels, kernel size 3, stride 1, padding 1 |
| 10 | Batch normalization layer |
| 11 | ReLU activation |
| 12 | Convolutional layer: 256 input channels, 256 output channels, kernel size 1, stride 2, padding 0 |
| 13 | Convolutional layer: 256 input channels, 512 output channels, kernel size 3, stride 1, padding 1 |
| 14 | Batch normalization layer |
| 15 | ReLU activation |
| 16 | Convolutional layer: 512 input channels, 512 output channels, kernel size 1, stride 2, padding 0 |
| 17 | Adaptive average pooling layer: reduces spatial dimensions to $1 \times 1$ |
| 18 | Flatten operation |
| 19 | Dropout layer: $p = 0.5$ |
| 20 | Linear layer: 512 input features to 10 output classes |

**OfficeHome**. For the OfficeHome dataset, we use OfficeHomeNet, a model based on a pretrained ResNet-18 architecture [63]. To adapt the model for the dataset's complexity, we replace the original fully connected layer with a custom classification head. We fine-tune the model by training only the final convolutional block (layer4) and our custom head, keeping all earlier layers frozen. This results in a model with 8,541,761 trainable parameters out of 11,324,545 total. The custom head consists of:

| Layer | Layer Description |
|---|---|
| 1 | Linear layer: 512 input features to 256 output features |
| 2 | ReLU activation |
| 3 | Dropout layer: $p = 0.5$ |
| 4 | Linear layer: 256 input features to 65 output classes |

**MEMD-ABSA**. For the MEMD-ABSA dataset, we fine-tune a model based on the TinyBert architecture[64], called TinyBertForSentiment. The model comprises 4,386,307 parameters, and has the following structure:

| Layer | Layer Description |
|-------|-------------------|
| 1 | Embeddings Layer: word embeddings (30522 vocab size, 128 dim), position embeddings (512 max sequence length, 128 dim), token type embeddings (2 types, 128 dim), layer norm, dropout layer ($p = 0.1$) |
| 2 | Encoder layer 1: transformer encoder block |
| 3 | Encoder layer 2: transformer encoder block |
| 4 | Pooler: linear layer (128 to 128), Tanh activation |
| 5 | Dropout layer: $p = 0.1$ |
| 6 | Linear layer: 128 input features to 3 output classes |

Each transformer encoder block is comprised of a 2-head self-attention mechanism with a hidden size of 128 and GELU activation functions. We tokenize input text using a pretrained BERT tokenizer that employs the WordPiece tokenization[65]. All sequences are subsequently padded or truncated to a fixed length of 128 tokens.

## E.3   Pretraining Phase

The pretraining phase begins by partitioning the entire dataset into training and holdout sets. We utilize the 80:20 split and uniform random sampling for the image datasets, and adopt the predefined training and validation (holdout) splits for the text dataset. We then filter both the training and holdout sets to contain only samples from a single designated source domain. The resulting single-domain training set is denoted by $\mathcal{D}$.

A new agent with a custom model is then instantiated and the model is trained on $\mathcal{D}$. Upon initialization of the loss criterion and optimizer, the training loop begins. Within each iteration, the agent employs the `update steps` method to refine the model parameters. Unlike traditional epoch-based training, this method samples batches, each of size $|\xi|$, from the training set for $n_{\text{steps}}$ steps. For each batch, it computes the gradients and updates the model with the specified optimizer and learning rate $\eta$. This process is repeated for a total of $T$ updates.

After each training iteration, the agent evaluates the revised model on the single-domain holdout set. For the image datasets, to prevent overfitting, we employ an early stopping mechanism that terminates the training process if the holdout set validation accuracy exceeds a predefined threshold $\text{acc}_{\text{thresh}}$. For the text dataset, we train for the full $T$ iterations. Upon completion, we save the parameters of the final model for the subsequent evaluation phase, which corresponds to the last updated model for the image datasets and the best model for the text dataset.

All hyperparameters are detailed in Table 5. Their values were determined through a combination of grid search and random search.

Table 5: Hyperparameter settings for the pretraining phase across PACS, DigitsDG, OfficeHome, and MEMD-ABSA datasets.

| Hyperparameter | PACS | DigitsDG | OfficeHome | MEMD-ABSA |
|----------------|------|----------|------------|-----------|
| Learning Rate ($\eta$) | 0.005 | 0.05 | 0.05 | 5e-5 |
| Batch Size ($|\xi|$) | 128 | 128 | 256 | 32 |
| # of Steps/Iter ($n_{\text{steps}}$) | 10 | 10 | 20 | 60 |
| Number of Iterations ($T$) | 200 | 200 | 100 | 10 |
| Max Sequence Length | N/A | N/A | N/A | 128 |
| Loss | Cross Entropy | Cross Entropy | Cross Entropy | Cross Entropy |
| Optimizer | SGD | SGD | SGD | AdamW |
| Momentum | 0.9 | 0.9 | 0.9 | N/A |
| $\text{acc}_{\text{thresh}}$ | 75% | 75% | 90% | N/A |

## E.4   Evaluation Phase

**Dataset and General Setup**. The initial data partitioning and single-domain filtering for the evaluation phase are identical to the pretraining phase. From these domain-specific splits, we construct a training set, with a fixed size $|\mathcal{D}_{\text{set}}|$, and a corresponding holdout set of size $0.25 \times |\mathcal{D}_{\text{set}}|$.

The size $|\mathcal{D}_{\text{set}}|$ is chosen to ensure that any single domain contains sufficient samples to fully populate these sets after the initial partitioning.

We then initialize the agent and run a concept drift simulation for $T = 250$ time steps. The drift is simulated by modeling a dynamic environment in which the data distributions change over time. At each time step $t$, the agent's model performance (e.g., loss and accuracy) is measured exclusively on the holdout set. This performance information guides the agent's update policy that determines whether to retrain the model. If an update is triggered, the agent retrains its model on the training set using the `update steps` method with $n_{\text{steps}}$ steps, a batch size of $|\xi|$, and a learning rate $\eta$. The policy operates under a budget constraint $\bar{\lambda}$, which, given a constant update cost $\lambda$, results in an upper bound on the effective update rate equal to $\frac{\bar{\lambda}}{\lambda}$. All evaluation phase hyperparameters are summarized in Table 6.

Table 6: Update hyperparameter settings for the evaluation phase across all datasets.

| Hyperparameter | PACS | DigitsDG | OfficeHome | MEMD-ABSA |
|---|---|---|---|---|
| Training Set Size ($|\mathcal{D}_{\text{set}}|$) | 1024 | 1024 | 2048 | 1024 |
| Learning Rate ($\eta$) | 0.01 | 0.05 | 0.05 | 5e-5 |
| Batch Size ($|\xi|$) | 128 | 128 | 256 | 32 |
| # of Steps/Update ($n_{\text{steps}}$) | 5 | 1 | 5 | 60 |
| Max Sequence Length | N/A | N/A | N/A | 128 |
| Loss | Cross Entropy | Cross Entropy | Cross Entropy | Cross Entropy |
| Optimizer | SGD | SGD | SGD | AdamW |

**Drift Schedules.** To simulate concept drift, we employ a *drift scheduler*. The scheduler alters the domain composition of the training and holdout sets by replacing existing samples with data from new target domains. For any given time $t$, this process is governed by a *drift schedule*, which defines the specific pattern, timing, and rate of this change, including periods of no drift where the rate is zero.

We implement a total of seven schedules: four are presented in the main paper, with three additional schedules introduced in Appendix F.3. All schedules use a `replace` strategy, where a fraction of the existing data is substituted with samples from the target domains. This ensures the dataset size remains constant throughout the simulation.

The specific configurations are detailed below using the PACS dataset as an example. The setups for DigitsDG, OfficeHome, and MEMD-ABSA are analogous, using their respective domains.

- **Burst**: Introduces new domains in periodic deterministic bursts. After an initial delay of 45 time steps, the first burst starts at $t = 45$ (introducing Photo domain data) and the second starts at $t = 165$ (introducing Cartoon). Each burst lasts for 3 time steps with a drift rate of 0.4, effectively replacing the entire dataset.

- **Step**: Introduces domains in discrete steps with increasing drift rates. After an initial period of no drift, the drift rate is set to 0.004 at $t = 60$ (introducing Sketch), changing to 0.006 at $t = 120$ (introducing Cartoon), and finally changing to 0.008 at $t = 180$ (introducing Art Painting).

- **Wave**: Consists of repeating cycles. After an initial delay of 50 time steps, a wave begins where the domains are introduced at a rate of 0.032 for 30 time steps. This is followed by a 70-time-step period of no drift. This 100-step cycle (30 drift and 70 no drift) then repeats. Domains cycle in the order: Photo, Cartoon, Sketch.

- **Spikes**: Extends the Burst schedule with randomized characteristics. The spike interval is uniformly random between 90 and 130 time steps. Similarly, the initial delay is between 30 and 60, the duration is between 3 and 6 time steps, and the rate is between 0.3 and 0.6. Domains cycle in the order: Photo, Cartoon, Sketch.

- **Constant**: The drift occurs at a constant rate of 0.016. The target domain cycles every 50 time steps in the order: Sketch, Photo, Cartoon, Art Painting.

- **Decaying Spikes**: Modifies the Burst schedule with increasing intervals between spikes. After an initial delay of 20 time steps, the first spike occurs. The interval to the next spike starts at 30 and

increases by 10 after each subsequent spike. Each spike lasts for 3 time steps and has a drift rate of 0.35. Domains cycle in the order: Sketch, Photo, Cartoon.

- **Seasonal Flux**: Simulates cyclic drift. After an initial delay of 10 time steps, the schedule cycles between the Photo and Sketch domains. A full cycle (e.g., from Photo to Sketch and back) takes 150 time steps. The drift rate varies sinusoidally between 0.001 and 0.016.

**Update Policies** We implement the proposed RCCDA policy and four baseline strategies, detailed below.

- **Uniform** (Policy 1): Retrains randomly with a fixed probability $\frac{\bar{\lambda}}{\lambda} = \bar{\pi}$.

- **Periodic** (Policy 2): Retrains at fixed intervals $\frac{\bar{\lambda}}{\lambda} = \frac{1}{\bar{\pi}}$

- **Budget-Increase** (Policy 3): Retrains when the loss increases for three consecutive time steps at the end of a sliding window of size $w = 40$, provided a sufficient budget is available. The budget accumulates at a rate of $\frac{\bar{\lambda}}{\lambda} = \bar{\pi}$ per time step.

- **Budget-Threshold** (Policy 4): Retrains when the current loss exceeds the maximum loss over a window of size $w = 40$ by a threshold $\epsilon = 0.1$, provided a sufficient budget is available. The budget accumulates at a rate of $\frac{\bar{\lambda}}{\lambda} = \bar{\pi}$ per time step.

- **RCCDA** (Policy 5): The proposed policy with estimation defined as $\hat{\mathcal{G}}(\mathcal{H}_t) = K_p(f(\theta_{t-1}, \xi_t) - \min_{i \in \{0,\ldots,t\}} f(\theta_i, \xi_i)) + K_d(f(\theta_i, \xi_i) - f(\theta_{t-1}, \xi_{t-1}))$. The update occurs if: $V K_p(f(\theta_{t-1}, \xi_t) - \min_{i \in \{0,\ldots,t\}} f(\theta_i, \xi_i)) + V K_d(f(\theta_i, \xi_i) - f(\theta_{t-1}, \xi_{t-1})) \geq Q(t) + 0.5 - \frac{\bar{\lambda}}{\lambda}$. We set $V = 10$ constant across all experiments, and tune the $K_p, K_d$ values for different settings and resource constraints. The virtual queue is updated as $Q(t + 1) = \max\left\{0, Q(t) + \pi(t) - \frac{\bar{\lambda}}{\lambda}\right\}$, where $\pi(t) \in \{0, 1\}$ is the update decision at time $t$.

In Tables 7 and 8 we summarize the values of $K_p, K_d$ used by RCCDA across the conducted experiments.

Table 7: Optimal $(K_p, K_d)$ parameters for RCCDA across various datasets and drift schedules. The update rate was set to 0.1 for the image datasets and 0.01 for MEMD-ABSA.

| Drift Schedule | PACS | DigitsDG | OfficeHome | MEMD-ABSA |
|---|---|---|---|---|
| Burst | $(1.0, 0.1)$ | $(2.5, 0.5)$ | $(1.0, 0.5)$ | $(0.1, 2.5)$ |
| Step | $(1.0, 0.1)$ | $(1.0, 0.5)$ | $(1.0, 0.5)$ | N/A |
| Wave | $(0.5, 0.1)$ | $(1.0, 0.5)$ | $(1.0, 0.5)$ | N/A |
| Spikes | $(0.75, 0.5)$ | $(2.0, 0.1)$ | $(1.0, 0.5)$ | N/A |
| Constant | $(0.2, 0.1)$ | $(0.3, 0.25)$ | N/A | N/A |
| Decaying Spikes | $(0.5, 0.1)$ | $(0.5, 0.1)$ | N/A | N/A |
| Seasonal Flux | $(0.3, 0.25)$ | $(0.5, 0.1)$ | N/A | N/A |

Table 8: $(K_p, K_d)$ parameters used by RCCDA across different update rates for the Burst drift schedule. Evaluated on the PACS and DigitsDG datasets

| Update Rate $\frac{\bar{\lambda}}{\lambda}$ | PACS | DigitsDG |
|---|---|---|
| 0.02 | $(0.06, 0.05)$ | $(0.1, 0.05)$ |
| 0.03 | $(0.13, 0.05)$ | $(0.28, 0.05)$ |
| 0.05 | $(0.5, 0.1)$ | $(0.55, 0.1)$ |
| 0.07 | $(0.7, 0.1)$ | $(1.0, 0.2)$ |
| 0.15 | $(4.0, 1.0)$ | $(4.0, 1.0)$ |
| 0.20 | $(7.0, 2.5)$ | $(7.0, 2.0)$ |
| 0.25 | $(10.0, 4.0)$ | $(14.0, 6.0)$ |
| 0.30 | $(14.0, 5.5)$ | $(20.0, 8.0)$ |

**Evaluation Loop** With this setup, we execute the evaluation loop for $T = 250$. At each time step $t$, the loop executes the following operations:

1. **Apply Drift**: The drift scheduler determines the current drift rate and target domains. Drift is then applied to the agent's training and holdout datasets.

2. **Evaluate Performance**: The agent's model performance (current loss and accuracy) is computed on the holdout set.

3. **Policy Decision**: The agent's policy determines whether to retrain based on current and historical performance metrics.

4. **Retrain if Decided**: If retraining is triggered, the agent updates its model parameters for $n_{\text{steps}}$ steps using the training set.

**Results** All results are averaged over multiple random seeds (10 for experiments in the main paper and 3 for those in the Appendix). In Figure 2, we consider a single starting domain for each of the drift schedules for the PACS dataset. For the numerical results in Table 1, we further average the seed-averaged results across different starting domains. Specifically, we average the results for PACS over starting domains: Photo, Art Painting, Cartoon, Sketch; for DigitsDG over starting domains: MNIST, MNIST-M, and SVHN; for OfficeHome over starting domains: RealWorld, Product, Art; and for MEMD-ABSA over starting domains: Books, Clothing, Hotel, Laptop and Restaurant. In all cases, the reported accuracy is the average performance on the holdout set over the entire operational period $T$.

# F   Additional Experimental Results

## F.1   OfficeHome Dataset

In this section, we provide additional results on the OfficeHome [56] dataset, a domain generalization benchmark consisting of four domains: Art, Clipart, Product, and Real-World.

The results are reported in Table 9. They are consistent with our findings on the PACS and DigitsDG datasets (Sec. 6), with RCCDA achieving significant performance improvements for the Burst schedule, moderate improvements for the Wave and Spikes schedules. On the Step schedule, it achieves the highest accuracy, performing competitively with the other baselines. This further demonstrates the effectiveness of our policy under resource constraints.

Table 9: Average validation accuracy (%) of policies across concept drift schedules for the OfficeHome dataset. The mean update rate constraint is $\frac{\bar{\lambda}}{\lambda} = 0.1$. The best-performing method for each configuration is shown in **bold**.

| Policy | OfficeHome | | | |
|---|---|---|---|---|
| **Drift Schedule** | Burst | Step | Wave | Spikes |
| **RCCDA (ours)** | **86.9** $\pm$ 2.8 | **90.0** $\pm$ 1.8 | **87.5** $\pm$ 2.9 | **89.9** $\pm$ 2.9 |
| Uniform | 81.9 $\pm$ 3.3 | 88.9 $\pm$ 2.5 | 85.0 $\pm$ 3.0 | 85.8 $\pm$ 5.2 |
| Periodic | 82.8 $\pm$ 3.5 | 89.6 $\pm$ 2.2 | 86.3 $\pm$ 3.0 | 88.3 $\pm$ 4.8 |
| Budget-Increase | 70.7 $\pm$ 4.7 | 89.3 $\pm$ 2.2 | 85.4 $\pm$ 3.2 | 79.3 $\pm$ 3.7 |
| Budget-Threshold | 77.7 $\pm$ 5.2 | 85.4 $\pm$ 2.6 | 79.2 $\pm$ 7.5 | 87.7 $\pm$ 2.4 |

## F.2   MEMD-ABSA Dataset - Textual Modality

In this section, we provide additional results on the MEMD-ABSA dataset [57], a multi-domain benchmark for textual Aspect-Based Sentiment Analysis (ABSA) with reviews from five domains: Books, Clothing, Hotel, Laptop and Restaurant. This experiment evaluates our method on a new, textual, modality, and explores its performance in a setting with less severe concept drift.

Our theoretical analysis is general and not limited to a specific data modality. Consequently, our policy is provably modality-invariant, with only its algorithmic constants being task-dependent. Operationally, however, the efficacy of RCCDA strongly relies on the magnitude of performance degradation detected during a distributional shift. On the MEMD-ABSA dataset, this degradation is modest when transferring between domains, especially compared to the vision tasks, as we demonstrate in Table 10. While the top-performing model for any given domain is the one trained specifically on it, the performance drop from using models trained on other domains is not significant,

with those models still achieving competitive classification accuracy. This strong generalization suggests that, for the architecture used, the textual domains are not sufficiently distinct to generate a concept drift as severe as that observed in our vision-based experiments.

Table 10: Cross-domain test accuracy (%) for TinyBERT on the MEMD-ABSA dataset. Each column shows the performance of models pretrained on different source domains and evaluated on the specified target domain.

| Pretrained on | Evaluated on | | | | |
|---|---|---|---|---|---|
| | **Books** | **Clothing** | **Hotel** | **Laptop** | **Restaurant** |
| **Books** | **81.7** $\pm$ 1.7 | 80.3 $\pm$ 0.6 | 88.2 $\pm$ 1.8 | 73.4 $\pm$ 1.6 | 73.6 $\pm$ 2.9 |
| **Clothing** | 75.0 $\pm$ 0.3 | **84.8** $\pm$ 0.6 | 88.8 $\pm$ 1.2 | 74.8 $\pm$ 1.6 | 74.5 $\pm$ 0.2 |
| **Hotel** | 72.0 $\pm$ 1.1 | 77.5 $\pm$ 0.9 | **95.8** $\pm$ 0.4 | 66.1 $\pm$ 1.8 | 72.6 $\pm$ 1.5 |
| **Laptop** | 70.7 $\pm$ 1.5 | 80.9 $\pm$ 1.2 | 84.2 $\pm$ 3.3 | **80.1** $\pm$ 0.2 | 73.8 $\pm$ 1.5 |
| **Restaurant** | 78.1 $\pm$ 1.4 | 81.3 $\pm$ 0.5 | 91.4 $\pm$ 1.4 | 75.6 $\pm$ 1.0 | **82.2** $\pm$ 0.3 |

Given this small inter-domain drift, we focused our evaluation on the Burst drift schedule, as it demonstrated the most pronounced performance differences among the policies. The results, summarized in Table 11, confirm that RCCDA still achieves the highest average accuracy, with a robust 1.4% improvement over the next-best performing baseline (Periodic). However, the performance gap is less pronounced than in the vision-based tasks, which is expected given the smaller effective drift magnitude. Notably, some baselines maintained high accuracy with a near-zero effective update rate (e.g. 0.001), indicating that in this low-drift setting, the model's inherent generalization ability was more influential than the update strategy. This result highlights that while RCCDA is robust across different modalities, the benefits of an intelligent update policy are most significant when the concept drift is severe.

Table 11: Average validation accuracy (%) of evaluated policies for the Burst drift schedule on the MEMD-ABSA dataset. The mean update rate constraint is $\frac{\bar{\lambda}}{\lambda} = 0.01$. The best-performing method is shown in **bold**.

| Policy | Average Validation Accuracy (%) |
|---|---|
| **RCCDA (Ours)** | **81.4** $\pm$ 2.9 |
| Uniform | 79.8 $\pm$ 3.5 |
| Periodic | 80.0 $\pm$ 3.2 |
| Budget-Increase | 79.2 $\pm$ 3.6 |
| Budget-Threshold | 79.2 $\pm$ 4.2 |

## F.3 Supplementary Concept Drift Schedules

In this section, we provide supplementary results on three additional drift schedules, designed to further evaluate the robustness of our proposed method, RCCDA. These schedules simulate diverse and challenging temporal distribution shifts:

(v) *Constant* - A steady drift where the target domain changes at a constant rate. The domains cycle every set number of steps.

(vi) *Decaying Spikes* - A series of sudden drift events, modifying the *Burst* schedule. After an initial delay, spikes occur at increasing intervals, becoming less frequent.

(vii) *Seasonal Flux* - A cyclical drift that simulates seasonal variations. The schedule alternates between two domains following a sinusoidal pattern with a constant period, amplitude, and offset.

Table 12 reports the average validation accuracy on the PACS and DigitsDG datasets for these new schedules. The results are consistent with our findings for drift schedules (i)-(iv) presented in the main paper, demonstrating that RCCDA consistently outperforms or matches all baseline methods across these new drift configurations.

Table 12: Average validation accuracy (%) of evaluated policies on the PACS and DigitsDG datasets for three additional concept drift schedules. The mean update rate constraint is $\bar{\lambda}/\lambda = 0.1$. The best-performing method for each configuration is shown in **bold**.

| Policy | PACS | | | Digits-DG | | |
|---|---|---|---|---|---|---|
| | Constant | Decaying Spikes | Seasonal Flux | Constant | Decaying Spikes | Seasonal Flux |
| **RCCDA (ours)** | **54.1** $\pm$ 1.5 | **59.7** $\pm$ 3.3 | **63.4** $\pm$ 2.8 | **58.1** $\pm$ 9.4 | **54.5** $\pm$ 15.4 | **63.7** $\pm$ 9.6 |
| Uniform | 51.3 $\pm$ 1.3 | 48.0 $\pm$ 5.9 | 55.1 $\pm$ 6.0 | 56.2 $\pm$ 7.2 | 48.0 $\pm$ 12.7 | 61.9 $\pm$ 8.2 |
| Periodic | 48.4 $\pm$ 3.5 | 51.3 $\pm$ 4.4 | 57.9 $\pm$ 5.8 | 56.8 $\pm$ 8.2 | 49.6 $\pm$ 14.0 | 62.0 $\pm$ 8.3 |
| Budget-Increase | 47.4 $\pm$ 2.4 | 42.1 $\pm$ 6.1 | 53.3 $\pm$ 4.3 | 55.6 $\pm$ 9.3 | 45.1 $\pm$ 16.8 | 61.4 $\pm$ 9.9 |
| Budget-Threshold | 45.5 $\pm$ 3.7 | 44.6 $\pm$ 10.9 | 58.2 $\pm$ 14.5 | 52.3 $\pm$ 6.7 | 49.0 $\pm$ 14.8 | 61.5 $\pm$ 11.5 |

## F.4 Varying Update Rate

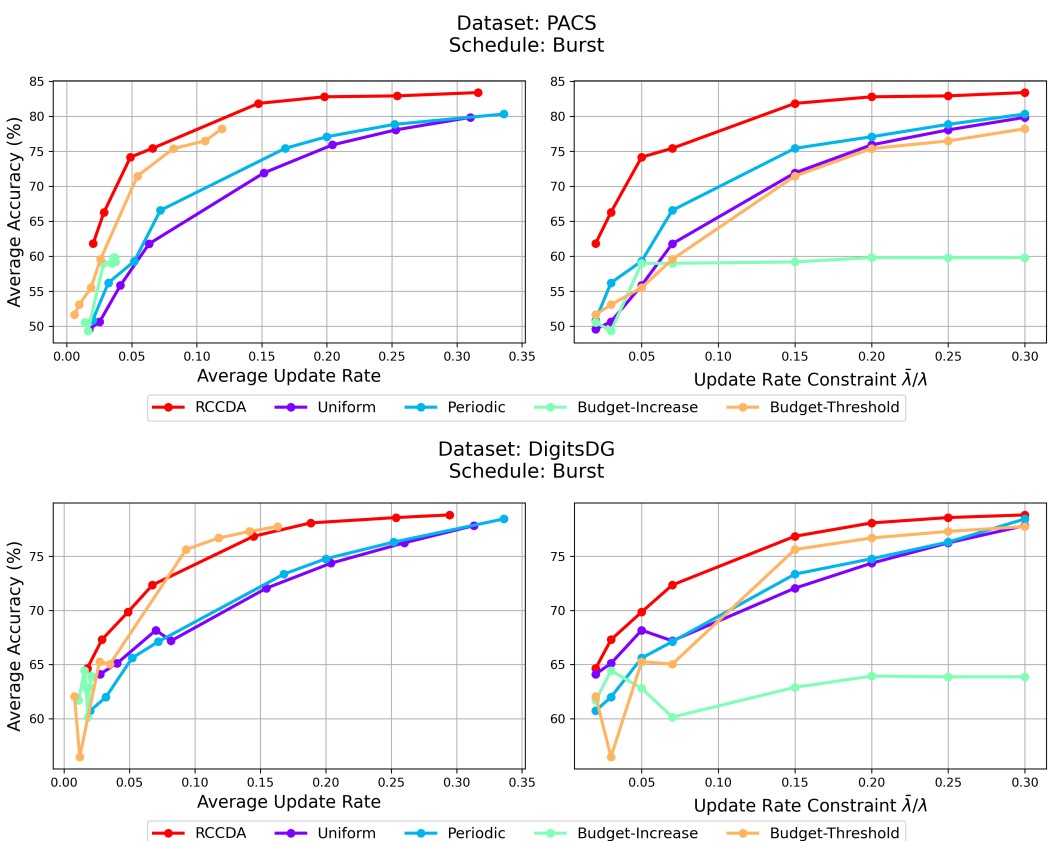

Figure 3: Average validation accuracy of policies across different update rates for the burst drift schedule, across PACS and DigitsDG datasets. The left column is the accuracy per real update rate, and the right column is the accuracy per constrained update rate $\frac{\bar{\lambda}}{\lambda}$.

In this section, we analyze the performance of RCCDA and baseline policies under various update rate constraints $\frac{\bar{\lambda}}{\lambda}$. The results for the Burst drift schedule om the PACS and DigitsDG datasets are available in Figure 3. This study yields several key insights into the behavior of our proposed policy.

First, the results confirm that RCCDA effectively adheres to the specified resource constraints. As demonstrated by comparing the left (effective update rate) and right (constrained update rate) columns of Figure 3, the average update rate achieved by our policy closely matches the predefined constraint $\frac{\bar{\lambda}}{\lambda}$. This indicates that our policy utilizes its allocated budget more effectively, making update decisions that lead to better model performance.

Notably, the Budget-Threshold policy was able to marginally surpass RCCDA in accuracy for effective update rates between 0.10 and 0.15 on the DigitsDG dataset. However, this observation highlights a critical flaw in the baseline policy design: a lack of precise resource utilization control. To achieve the effective update rate in this narrow range, the Budget-Threshold policy required a significantly higher allowed resource constraint. Essentially, it only achieved this rate because its update rule failed to spend the allocated (higher) budget, meaning it was operating under a much looser resource constraint.

In contrast, RCCDA's performance is achieved while strictly adhering to the specified time-average resource constraint. As such, the baseline's slight advantage is a result of its uncontrolled behavior; it adhered to a higher $\frac{\bar{\lambda}}{\lambda}$ constraint to achieve the same effective update rate. RCCDA's performance, while slightly lower, is a more accurate reflection of its ability to maximize accuracy under a true, enforced resource constraint.

### F.5 Robustness under Uncertainty

Our framework assumes that the exact model inference loss, $f_t$, can be accessed by our policy at any time step $t$. While this is a standard assumption in supervised drift adaptation literature, practical scenarios, especially ones involving small or partially labeled datasets, may only provide a high-variance, noisy estimate of the true loss. To evaluate the robustness of RCCDA under such conditions, we conducted an additional experiment quantifying the impact of a noisy loss estimate on the policy's performance.

The experiment was performed on the PACS dataset using the Burst drift schedule, with all hyperparameters and the environment identical to our main experiments (see Appendix E). The key modification was the introduction of synthetic noise to the loss signal accessed by the policy. Specifically, at each time step $t$, the policy received a noisy loss estimate, $f_t^{\text{estimate}}$, defined as:

$$f_t^{\text{estimate}} = f_t + \mathcal{N}\left(0, (k_u \times f_t)^2\right) \tag{34}$$

where the additive noise is sampled from a zero-mean normal distribution $\mathcal{N}$, with its standard deviation proportional to the true loss $f_t$ via the uncertainty coefficient $k_u$. We evaluated the policy's performance across several values of $k_u$. For each degree of uncertainty, we re-tuned the policy's $(K_p, K_d)$ parameters to ensure the effective update rate remained close to the target of $\frac{\bar{\lambda}}{\lambda} = 0.1$.

Table 13: Performance of RCCDA on the PACS dataset given (Burst schedule) under varying degrees of loss uncertainty.

| Degree of Uncertainty ($k_u$) | Validation Accuracy (%) | Effective Update Rate | $(K_p, K_d)$ |
|---|---|---|---|
| 0.0 (certain) | $79.4 \pm 5.2$ | $0.093 \pm 0.030$ | $(1.5, 0.3)$ |
| 0.05 | $79.3 \pm 6.1$ | $0.099 \pm 0.011$ | $(0.7, 0.3)$ |
| 0.1 | $79.2 \pm 6.8$ | $0.103 \pm 0.009$ | $(0.6, 0.25)$ |
| 0.2 | $78.0 \pm 7.0$ | $0.103 \pm 0.007$ | $(0.4, 0.25)$ |
| 0.3 | $77.4 \pm 8.0$ | $0.104 \pm 0.004$ | $(0.3, 0.25)$ |

The results, available in Table 13, demonstrate that RCCDA is highly robust to noisy loss signals. While the inference performance gradually degrades as the uncertainty coefficient $k_u$ increases, the drop is modest compared to the uncertainty increase. Even with significant uncertainty ($k_u = 0.3$), the validation accuracy decreases by only $2.0\%$ (a relative drop of $\sim 2.5\%$) compared to the certain case. At the same time, the standard deviation of the validation accuracy increases with higher uncertainty - an expected outcome given the noise introduces greater variability into the framework's behavior. Importantly, RCCDA successfully maintained an average effective update rate near the target $\frac{\bar{\lambda}}{\lambda} = 0.1$ across all uncertainty conditions, demonstrating its ability to effectively manage the resource budget is resilient to loss signal perturbations. This analysis validates that RCCDA is a practical solution for real-world scenarios where the loss signal may be imperfect.

### F.6 Replay Buffer

As established by our theoretical analysis, RCCDA is designed to optimize an immediate, per-time-step bound on convergence. This design prioritizes rapid, resource-efficient adaptation, making the

policy highly effective in environments characterized by new, rarely-recurring concept drifts. As a consequence of this focus, the policy is concentrated on adapting the model to the most recent data distributions, which can lead to the gradual forgetting of past data representations. While this objective is distinct from the explicit knowledge-retention goals of continual learning (see Section 2), it can pose a practical challenge in real-world scenarios with recurring drifts, such as those with seasonal or cyclical environmental changes, making recurring drifts a key consideration.

Table 14: Performance comparison of RCCDA with and without a replay buffer on a recurring drift schedule. Accuracy (%) is reported for each domain at specific time steps.

| Time Step | Domain | No Buffer (100% New) | With Buffer (80% New, 20% Old) |
|---|---|---|---|
| 50 | photo | $30.60 \pm 9.58$ | $30.60 \pm 9.58$ |
| | art painting | $21.35 \pm 2.31$ | $21.35 \pm 2.31$ |
| | cartoon | $80.08 \pm 11.16$ | $80.08 \pm 11.16$ |
| | sketch | $26.43 \pm 3.99$ | $26.43 \pm 3.99$ |
| 150 | photo | $78.26 \pm 3.26$ | $71.74 \pm 4.06$ |
| | art painting | $25.00 \pm 1.99$ | $27.60 \pm 1.92$ |
| | cartoon | $20.31 \pm 3.04$ | $63.93 \pm 2.71$ |
| | sketch | $18.49 \pm 0.80$ | $16.67 \pm 6.63$ |
| 250 | photo | $50.78 \pm 2.76$ | $57.29 \pm 4.90$ |
| | art painting | $63.28 \pm 3.31$ | $57.03 \pm 3.62$ |
| | cartoon | $36.33 \pm 3.38$ | $60.16 \pm 3.76$ |
| | sketch | $22.01 \pm 2.89$ | $21.88 \pm 2.41$ |
| 350 | photo | $39.71 \pm 4.61$ | $58.07 \pm 0.49$ |
| | art painting | $34.64 \pm 3.81$ | $43.75 \pm 3.31$ |
| | cartoon | $70.31 \pm 3.68$ | $79.82 \pm 5.94$ |
| | sketch | $30.60 \pm 2.89$ | $31.90 \pm 1.92$ |
| 450 | photo | $12.76 \pm 3.41$ | $48.44 \pm 7.10$ |
| | art painting | $15.10 \pm 4.47$ | $38.93 \pm 6.33$ |
| | cartoon | $28.39 \pm 16.44$ | $70.18 \pm 6.20$ |
| | sketch | $58.07 \pm 2.08$ | $57.94 \pm 2.96$ |
| 550 | photo | $79.43 \pm 0.80$ | $79.04 \pm 3.10$ |
| | art painting | $29.43 \pm 2.71$ | $41.41 \pm 4.82$ |
| | cartoon | $14.45 \pm 6.37$ | $65.10 \pm 7.91$ |
| | sketch | $18.49 \pm 0.80$ | $44.66 \pm 10.80$ |
| 650 | photo | $52.08 \pm 2.17$ | $53.52 \pm 10.73$ |
| | art painting | $76.17 \pm 10.13$ | $58.59 \pm 10.46$ |
| | cartoon | $38.28 \pm 2.73$ | $62.50 \pm 6.99$ |
| | sketch | $29.30 \pm 4.07$ | $46.88 \pm 6.27$ |
| 750 | photo | $42.84 \pm 6.95$ | $57.16 \pm 6.10$ |
| | art painting | $37.37 \pm 2.31$ | $52.21 \pm 6.20$ |
| | cartoon | $73.83 \pm 5.37$ | $75.26 \pm 8.60$ |
| | sketch | $30.86 \pm 6.51$ | $47.53 \pm 5.12$ |
| 850 | photo | $11.72 \pm 3.08$ | $55.08 \pm 3.08$ |
| | art painting | $12.24 \pm 1.44$ | $39.58 \pm 1.76$ |
| | cartoon | $15.89 \pm 5.09$ | $65.62 \pm 4.98$ |
| | sketch | $64.58 \pm 7.79$ | $62.50 \pm 12.77$ |
| 950 | photo | $80.86 \pm 4.47$ | $85.29 \pm 6.22$ |
| | art painting | $36.72 \pm 1.94$ | $49.35 \pm 7.56$ |
| | cartoon | $23.57 \pm 4.31$ | $61.33 \pm 9.13$ |
| | sketch | $18.62 \pm 0.97$ | $57.03 \pm 7.94$ |

To enhance long-term knowledge retention and adapt RCCDA for such environments, the framework can be naturally extended with a replay buffer mechanism. The buffer stores a small, representative subset of data from previously encountered distributions. When the policy triggers a model update, the training batch is then composed of a majority of new data alongside a minority of data sampled from this buffer. By systematically re-exposing the model to past concepts, this approach mitigates the effects of catastrophic forgetting and preserves knowledge across recurring domains.

To evaluate this extension, we conducted a preliminary experiment using a recurring burst schedule on the PACS dataset, where the active domain was cycled every 100 time steps (starting with Cartoon, then rotating Photo, Art Painting, Cartoon, Sketch in order). We compare two configurations: a baseline "No Buffer" agent, where the training set shifts entirely to the new domain at each drift event, and a "Buffer" agent. For the latter, at each drift event, the training set is recomposed such that 80% of the data is from the new target domain, while the remaining 20% forms a replay buffer. This buffer is populated by sampling uniformly from all previously-seen majority domains. Note that we utilized a fixed buffer size and a simple sampling strategy, as this experiment represents a preliminary investigation. A detailed exploration of buffer management presents a compelling direction for future work.

The results, presented in Table 14, show the model's accuracy an each domain at key time steps just before the next drift occurs. These findings reveal a clear trade-off between immediate adaptation and long-term knowledge retention. The "No Buffer" agent achieves a higher peak accuracy when a new target domain is introduced for the first time, such as for Photo at $t = 150$ (78.26% vs 71.74%) or Art Painting at $t = 250$ (63.28% vs 57.03%), as it dedicates all training resources to the new concept. In contrast, while the "Buffer" agent has a lower accuracy when a new domain is introduced, it mitigates catastrophic forgetting by maintaining a significantly higher accuracy on previously seen domains. For example, at $t = 150$, it achieves accuracy of 63.93% on the previous Cartoon domain, compared to just 20.31% achieved by the "No Buffer" agent. In addition, this retained knowledge proves advantageous when certain domains reappear, as evident with Cartoon at $t = 350$, where the "Buffer" agent achieves 79.82% validation accuracy, significantly higher than the 70.31% achieved by the "No Buffer" agent. This highlights a fundamental trade-off between maximizing immediate performance on novel drifts and ensuring robust generalization across recurring domains, and offers an interesting direction for future research.

## F.7 Statistical Significance of the Results

As mentioned in Appendix E, the main result presented in Table 1 are averaged not only across multiple random seeds but also across different starting domains for each dataset. While this provides a high-level summary, it can also introduce high reported standard deviation, as the policy performance over time will depend on the specific sequence of domains encountered. This variance can obscure the statistical significance of performance differences between policies.

To provide a more statistically robust view, we included a deconstructed analysis of the results. Table 15 breaks down the performance of all policies on the PACS dataset for the Burst drift schedule, showing the average validation accuracy for each starting domain individually. The results are averaged over 10 random seeds.

Table 15: Deconstructed validation accuracy (%) on the PACS dataset for the Burst schedule, by starting domain.

| Policy | Starting Domain | | | |
|---|---|---|---|---|
| | **Photo** | **Cartoon** | **Art Painting** | **Sketch** |
| **RCCDA (ours)** | **80.6** $\pm$ 7.4 | **85.8** $\pm$ 2.1 | **73.8** $\pm$ 5.3 | **71.7** $\pm$ 1.6 |
| Uniform | 78.0 $\pm$ 5.0 | 70.0 $\pm$ 1.8 | 67.6 $\pm$ 6.3 | 51.7 $\pm$ 6.8 |
| Periodic | 77.1 $\pm$ 4.3 | 70.9 $\pm$ 1.7 | 69.7 $\pm$ 5.5 | 57.4 $\pm$ 1.8 |
| Budget-Increase | 72.0 $\pm$ 5.8 | 61.3 $\pm$ 5.6 | 66.5 $\pm$ 1.4 | 36.7 $\pm$ 5.6 |
| Budget-Threshold | 79.4 $\pm$ 6.4 | 79.3 $\pm$ 2.1 | 67.8 $\pm$ 3.4 | 43.0 $\pm$ 5.0 |

This per-domain analysis provides further insight into the aggregated results. The high standard deviations reported in the main result can be attributed to high performance variations across the different source domains. The updated standard deviations are lower in most cases compared to the original result, with a few exceptions. Additionally, RCCDA again consistently outperforms all baseline policies in each source domain, with statistically significant improvement achieved for all domains except Photo. Notably, while original results suggested a potential performance overlap between RCCDA and the Budget-Threshold policy, this breakdown reveals that RCCDA achieves a higher mean accuracy in all cases, with the improvement being statistically significant in all domains except for Photo. As such, this analysis explains the high variance in the main results and confirms the statistical significance of our method's improvements.

## G    Code Documentation

In this section, we provide specific details on the code and hardware used for our experiments.

**Code Description** We wrote our code in Python 3.12.3, and utilized the following key imported modules:

- PyTorch (2.3.0)
- torchvision (0.18.0)
- numpy (1.26.4)
- `flwr_datasets` [54]
- PIL (from Pillow 10.3.0)
- pandas (2.2.3)
- matplotlib (3.9.2)
- seaborn (0.13.2)
- transformers (via huggingface-hub 0.26.3; for BertForSequenceClassification and BertTokenizer in MEMD-ABSA)
- standard libraries such as: os, argparse, json, time, random, etc.

We implemented a custom package to streamline code operations. The package code is available under `Federated_Learning_Base_Toolkit_torch`, and includes source code for our core classes and methods, such as `BaseNeuralNetwork` with the `update steps` method, `DriftAgent`, `class DomainDrift`, or the custom datahandlers and datasets. The code in `concept_drift_optimal_adaptation` uses these methods to implement pre-training, drift scheduling, update policies, and the policy evaluation code. We included the source code in the supplemental materials. Further details are available in our implementation.

**Executing Code** These instructions assume a Unix-like system (Linux/macOS); for Windows, use adjusted paths. The code can be executed through the following steps:

1. Assuming conda is installed, navigate to the directory with the code and run the commands:

   ```
   conda env create -f cog_fl_llm_env.yml
   ```

   ```
   conda activate cog_fl_llm_env
   ```

2. Navigate to directory `Federated_Learning_Base_Toolkit_torch`, and run the package installation command:

   ```
   python -m pip install .
   ```

3. To pretrain a model navigate to a dataset-specific directory, and run:

   ```
   python3 pretrain_models_<dataset>.py --options
   ```

   where `<dataset>` is the name of the corresponding dataset (PACS, DigitsDG, OfficeHome, or MEMD-ABSA), and `options` specify various parameters of the pretraining, the details are available in the code.

   - PACS loads via `fl_toolkit/PACSDataHandler` (`torchvision.datasets.PACS` with `download=True`); no manual download or `root_dir` needed.
   - For DigitsDG, OfficeHome, and MEMD-ABSA, download the dataset from source and update `root_dir` in code.

4. To evaluate a model, navigate to a dataset-specific directory, and run

   ```
   python3 evaluate_policy_<dataset>.py --options
   ```

   where `options` specify various parameters of the evaluation process, the policy evaluated, the setting used, and the drift schedule.

   - Uses the models from the pretraining step.

- For DigitsDG, OfficeHome, and MEMD-ABSA datasets, a root directory with the downloaded dataset has to be specified in order for the code to work.

Our entire codebase is available at:
`https://github.com/Adampi210/RCCDA_resource_constrained_concept_drift_adaptation_code`.

**Hardware**. For each experiment, we utilized 14 Intel Xeon Platinum 8480+ CPU cores and 1 NVIDIA H100 GPU. CPU memory was allocated proportionally to the core request, with approximately 9 GB per core, yielding about 126 GB of CPU memory per job. The H100 GPU provided 80 GB of dedicated memory. The GPU executed all seeds concurrently for a given experimental configuration. The upper time limit for pretraining was set to 2 days, and for each evaluation job was set to 6 hours. In practice, the pretraining took at most 2 hours per configuration. Notably, the evaluation execution time varied by dataset, with the DigitsDG dataset taking 5 minutes per configuration on average, PACS evaluations averaging 15 minutes (up to 20 minutes), and OfficeHome evaluations averaging around 30 minutes per configuration.

