# OpenReview forum: "RCCDA: Adaptive Model Updates in the Presence of Concept Drift under a Constrained Resource Budget"
_NeurIPS.cc/2025/Conference — NeurIPS 2025 poster_

### Official Review · Reviewer_8AhD · 2025-06-07

**Clarity:** 3
**Significance:** 2
**Originality:** 3
**Rating:** 4
**Confidence:** 4

**Summary:**

The work proposes a concept drift adaptation method which is temporally resource-aware. It works on the principle of deciding whether to update the model with the current batch based on the expected loss change, which in turn is determined by running inference on the current batch. The theoretical analysis derives convergence and stability bounds, showing the viability of the proposed approach.

**Questions:**

**Questions**

a. Based on Line 45, can the authors support the claim of their policy being lightweight? Is it in terms of time?

b. What happens when there's no drift? Does the algorithm perform at least as well as standard gradient descent with non-adaptive or adaptive optimizer?

c. Similar to the above question, according to Algorithm 1; if there are no drifts, would there be no updates happening to the model? (Referring to lines 165, 166 "dynamically determines when to update the model based on historical performance metrics")

d. What are the practical scenarios where we can see these drifts for a single client or a user?

e. How realistic is the assumption of concept drift being bounded by the KL divergence between the past and the current distributions?

f. Can the authors please justify line 144: "For example, trying to detect the concept drift requires performing operations on the dataset, which is highly resource consuming"? And by "resource", do the authors mean runtime to reach convergence?

g. Why doesn't the "model related loss difference" part of Equation 6 have the same dataset? Why do we need both past and current datasets?

h. What is the storage overhead of storing both past and current data for Equation 6?

i. What is the memory overhead of persisting all the past gradients according to Algorithm 1?
___
**Minor Comments**

a. Till page 4, the definition of "resources" was not clearly given. Maybe the authors can explain from the beginning what they mean by resources, so the readers can align their expectations properly.

b. Avoid framing the research question (line 45) in can/does form, as the answer to such questions can be given in yes/no, which does not give further insights. A research question starting with what/why/how/under which conditions is generally preferred.

c. Line 153 is introducing some symbols before defining them.

d. Constants of Algorithm 1 "Inputs" should also be defined explicitly.

**Ethical Concerns:**

["NO or VERY MINOR ethics concerns only"]

**Final Justification:**

While they support the claim that RCCDA is adaptable to resource-constrained settings, I remain somewhat skeptical about the practicality of the proposed use case. Changes in lighting, seasons, or weather can often be categorized as recurring drifts, which would require RCCDA to perform additional work due to its tendency to "forget" past distributions. Given these considerations, I believe a borderline accept is an appropriate evaluation.

**Limitations:**

yes

**Quality:**

2

**Strengths And Weaknesses:**

**Strengths**

The manuscript is well-written. The proposed method is novel and explained thoroughly with rationale behind each step.
____

**Weaknesses**

a. The motivation behind using "time" as the only resource is unclear. In my opinion, peak memory consumption or computation cost (per-iteration and total) are also good candidates for measuring the resource usage. Furthermore, if the authors are targeting time as their main resource, then Figure 2 needs to be in wall-clock time instead of number of steps for a fairer comparison.

b. Computation and memory overheads of the proposed approach versus its baselines are not given, making it hard to gauge the feasibility and adaptability of the approach. E.g., the virtual queue is tracking all the past gradients, that must cause a linear increase in memory per step. Furthermore, it's unclear if the inference time before each training iteration is included in Figure 2. A more explicit breakdown of (a) inference time before each training iteration, and (b) training time itself would be interesting to see.

c. While the related work section is well-written, line 88 stating "these approaches demand ... continuous data monitoring and frequent model updates, making them impractical for resource-constrained settings" seems similar to what the proposed approach is also doing. At each iteration, the historical gradients and losses (as well as current losses) are used to determine whether to update the model. Isn't that also continuous monitoring? And doesn't that add a temporal overhead?

d. Since the work only focuses on optimizing for the current data distribution, when the older data distributions reemerges, the algorithm would have to do more work (for retraining) to recover the previously trained weights. It's unclear under what scenarios or use cases we would only have to care about the current distribution. In this regard, continuous learning methods seem more flexible.

e. The results of Table 1 do not look statistically significant for the proposed approach, RCCDA. As an example, for PACS dataset and Burst drift case, Budget-Threshold has average accuracy of 67.4% with the deviation of +/- 14.9%, making it as high as 82.3, while RCCDA can only go as high as 77.3%. Similar observations can be made for other datasets as well.

---

> ### Author Rebuttal · Authors · 2025-07-28
>
> We thank the reviewer for the insightful comments and constructive feedback. We address the specific weakness and questions below and hope that our clarifications and planned revisions will resolve the concerns raised.
>
> ## Time as the only resource
> Our **theoretical framework is designed to be general and is not limited to "time" as a resource**. In the evaluated scenario, a model update is a discrete event represented by a general time-varying function, $\lambda(t)$. This function is an abstraction that can represent various types of resources, including peak memory consumption or computation cost. Similarly, the resource constraint $\bar \lambda$ is an average cost constraint over this general function $\lambda(t)$. For our experiments, we chose $\lambda(t)$ to be a constant-update computation cost, which makes the resource constraint equivalent to limiting the frequency of updates, effectively framing time as the only resource. We did this as it allows us to compare performance differences between policies as a result of their update timing decisions. Furthermore, such setup avoids the complexities of precisely simulating hardware-dependent metrics, which can vary significantly based on the underlying system. As such, Figure 2 uses "Time step" as the x-axis because it represents the discrete decision to spend resources at a given time in our problem formulation, making it a fair comparison between the evaluated policies.
>
> ## Computation and memory overheads of RCCDA
> The **main computational and memory bottlenecks in RCCDA are associated with the gradient estimation function** $\hat g(\cdot)$ and the history buffers $\mathcal H$. The choice of the estimation function is left to the agent, allowing for flexibility given memory and time constraints. In our experimental section, we utilized gradient estimation function $\hat{g}(\cdots) = K_{d} (f_t-f_{t-1})$, as per line 315 in the manuscript. This estimation function only requires storing the previous loss value, resulting in **minimal increase in memory overhead**. All other parts of our proposed algorithm are either computed as part of the operation of the system, such as $f_t$ (independent of the decision process), or take minimal computation time. Additionally, the **virtual queue** keeps track of a single scalar value and only requires a single update at any time $t$, keeping the **added memory and computational overhead to a minimum**. Overall, compared to the introduced baselines, our algorithm  performs a few more constant-time operations on average, but these add negligible computational and memory cost.
>
> Regarding the inference time, in our framework, we model the agent as being deployed in the evaluated environment. As such, the agent runs the evaluation task at every time $t$, modeling continuous operation, and making the inference time independent of the decision process. Similarly, when it comes to the training time, that time is captured implicitly in the abstract model update cost $\lambda(t)$.
>
> ## Continual data monitoring
> Thank you for pointing out this opportunity to improve the presentation. To clarify, what distinguishes RCCDA from the existing baselines is the cost associated with the monitoring operation. Traditional TDG or drift detection methods often require monitoring the data distribution properties, or sequentially evaluating all datapoints. In contrast, **RCCDA monitors only the model's inference loss**. The loss is a lightweight, scalar value that can be computed by the evaluation process of the model's primary task. Its computation is parallelizable and is optimized through a variety of techniques. Reliance on loss makes our approach lightweight, and as discussed before, results in minimal added temporal overhead.
>
> To illustrate this point, during the rebuttal, **we conducted additional experiment that compares the operation time of RCCDA with two established drift detection methods**: ADWIN, which monitors the model error rate, and Kolmogorov–Smirnov (K-S) test, which monitors the incoming data stream. We measured the clock-time taken by each method to finish detecting drift/making a decision after the initial model evaluation. We utilized the PACS dataset and a single Nvidia H100 GPU, and averaged the results over 4 different domains and 3 different seeds.  The results are:
>
> |Algorithm|Additional Decision Making Time (ms)| Relative Slowdown (vs. RCCDA) |
> |-|-|-|
> |RCCDA|0.000744± 0.000142| 1x|
> |ADWIN|1.845618 ± 0.469732|~2480|
> |KS-Test|125996.5 ± 9605.7| ~169,350,134|
>
> Clearly, **the temporal overhead added by our method is orders of magnitude smaller** compared to the established drift detection methods.
>
> ## Older data distributions reemerging
> Accounting for reemergence of older data distributions is an important consideration in Continual Learning, where the goal is long-term knowledge retention. **RCCDA, by contrast, is designed for scenarios where adapting to the most recent data distribution is the primary objective**. The goal of our work is immediate resource-efficient adaptation to a drifting distribution of a single task. This is an altogether different problem formulation from continual learning, with a different set of objective, and results in natural a trade-offs between the model's in-time performance and generalizability.
>
> ## Results statistical significance
> The high standard deviations of the results stem from averaging the result across different starting domains, which have highly varying performances. To illustrate this point, we can further break down the average validation accuracy and its standard deviation for all policies across the different starting domains. The result shown below is for the burst drift schedule on the PACS dataset for the update rate of $0.1$ (and using 10 seeds instead of 3):
>
> ||Photo|Cartoon|Art Painting|Sketch|
> |-|-|-|-|-|
> |RCCDA|80.6%±7.4%|85.8%±2.1%|73.8%±5.3%|71.7%±1.6%|
> |Uniform|78.0%±5.0%|70.0%±1.8%|67.6%±6.3%|51.7%±6.8%|
> |Periodic|77.1%±4.3%|70.9%±1.7%|69.7%±5.5%|57.4%±1.8%|
> |Budget-Increase|72.0%±5.8%|61.3%±5.6%|66.5%±1.4%|36.7%±5.6%|
> |Budget-Threshold|79.4%±6.4%|79.3%±2.1%|67.8%±3.4%|43.0%±5.0%|
>
> When deconstructed in this way, our policy consistently outperforms the baseline methods, demonstrating that the high standard deviation stems from averaging over the different starting domains. This result is also true for other drift schedules and datasets, which we will include in the final version of the manuscript.
>
> ## Response to questions
> a. **Lightweight claim**: As demonstrated in our earlier response, the added computation overheads of RCCDA are minimal, supporting the claim of our policy being lightweight.
>
> b & c. **Behavior with no drift**: Under no drift, the loss $f_t$ remains stable, with in the term $(f_t - \min_i f_i)$ approaching zero. As such, the likelihood of an update becomes small, diminishing the value of $Q(t)$ over time, leading to a stop in model updates, thus conserving resources. This is a key feature of RCCDA: the algorithm improves over standard gradient descent methods by not diminishing resources on unnecessary updates in low-drift environments.
>
> d. **Practical scenarios for drift**: Drifts are common in single user scenarios like adapting to a user's changing interests in recommender systems, mobile devices adapting to a changing environments (lighting, seasons, weather, etc.), or real-time systems like ad prediction.
>
> e. **Realism of bounded KL-divergence**: This is a standard theoretical assumption in the analysis of concept drift. It formalizes the notion that the data distribution will not change arbitrarily quickly between two consecutive time steps in practice.
>
> f. **Clarification of line 144**: As seen in the new results above, drift detection operations that require data monitoring like the K-S test take significantly longer than RCCDA. On the other hand, ADWIN, by sequentially comparing model outputs with targets for drift detection, is order of magnitudes slower than the proposed RCCDA algorithm. By "resource" in this context we mean the number of operations required to determine whether a drift happened or to make a policy decision.
>
> g. **Different datasets in Eq 6**: Equation 6 represents the objective of minimizing the time-averaged loss increase, which is then used in Lyapunov optimization to derive our update policy. Line 631 in the manuscript clarifies that the model difference arises from unrolling the telescoped loss $\mathbb E[f(\theta_0, \mathcal D_0)] - \mathbb E[f(\theta_T, \mathcal D_T)]$. The result is then combined with the drift-induced loss to define the Lyapunov optimization objective. The loss terms themselves come from the derived convergence bound, and the Lyapunov optimization goal is to find an approximate solution minimizing it.
>
> h. **Storage overhead of Eq 6**: Equation 6 is part of the problem formulation but not the algorithm. The final practical implementation (Algorithm 1 block) does not require storing past and current full datasets. At any time $t$, the agent only needs to store the current dataset $\mathcal D_t$ reflecting the current data distribution.
>
> i. **Memory overhead of storing past gradients**: As mentioned earlier, the memory overhead associated with storing the historical data of the gradients, loss values, and policy decisions depends on the gradient estimator function. In the practical algorithm implementation for our experiments, that overhead was minimal and reduced to storing just one additional scalar. The history buffers are meant to allow the agent to decide on how accurately they want to estimate the gradient magnitude versus how much computation and time they want to devote to that improving estimation accuracy.
>
> We will address the minor comments in the final version as well.
>
> Once again, thank you for your detailed comments! We will add all of these responses to the final version. Please let us know if there are any other questions we can answer.

---

> > ### Comment · Reviewer_8AhD · 2025-08-03
> >
> > Thank you for the response. I am still concerned about the following:
> >
> > 1. I disagree with the authors only considering time as a resource in a “resource-constrained” setting in this work. Either just mention “time-constrained” settings or give quantifiable results on memory and compute costs. In their rebuttal, the mentions of memory and compute costs as “minimal” or “negligible” are non-scientific, giving a reader no additional knowledge of the practicality of their proposed algorithm.
> >
> > 2. The approaches used to show effectiveness of the algorithm in terms of additional decision making time (ADWIN and Kolmogorov–Smirnov test) were not considered as baselines for the main experiments in the paper. It would have been informative to know the difference in accuracy for of RCCDA against the aforementioned baselines.
> >
> > 3. Under no drift, the likelihood of a gradient update becomes small. I would expect a drift detection algorithm to at least work as good as the more prevalent adaptive optimizer based algorithms, without any drift.
> >
> > 4. There are no citations provided for the claim about “concept drift being bounded by the KL divergence” being a standard assumption.
> >
> > 5. I disagree with the authors calling RCCDA “lightweight” until I see a holistic comparison of resource (Memory and Cost) consumption across the baselines.

---

> > > ### Author Response · Authors · 2025-08-04
> > > **Response to the Raised Questions**
> > >
> > > Thank you for additional feedback. We address each point below.
> > >
> > > Q1: Our framework is designed to be general and is not constrained to "time" as the only resource. It considers an agent operating over a time horizon $\mathcal T$ in a changing environment. At each time step $t$, the agent can perform a model update, which incurs a general, time-varying resource cost $\lambda(t)$. This cost is an abstraction that can represent any quantifiable resource, (e.g., FLOPs, energy, wall-clock time, etc). The agent's operational constraint is a total resource budget of $\bar \lambda T$ for the entire time horizon $\mathcal T$, where $\bar \lambda$ is the time-average resource constraint, used to bound resource usage in Equation (5): $ \frac{1}{T}\sum_{t=0}^{T-1} \lambda(t) \pi(t) \leq \bar{\lambda}$. This is equivalent to ensuring the total amount of consumed resources, $\frac{1}{T}\sum_{t=0}^{T-1} \lambda(t) \pi(t)$, does not exceed the available resource budget, $\bar \lambda T$.
> > >
> > > To show that this approach is generalizable, we added a new empirical analysis that quantifies the average resource usage and the resource constraint for several distinct resource types. We used the same settings as our main experiment on the PACS dataset, averaging the result over 250 updates. In our new analysis, we measure the resource usage for a model update for the following resource metrics: computation (GFLOPs), total data processed (GB), memory-time usage (GB-s), energy consumed (Joules), and wall clock time taken (s). The table below includes the results for the considered metrics, detailing the average cost per update ($\lambda$) for a given metric, and the total available resource budget ($\bar \lambda T$, T=250).
> > >
> > > |Resource Metric Type (Unit)|Constant Average Update Cost $\lambda$|Total Available Budget $\bar \lambda T$|
> > > |-|-|-|
> > > |Computation (GFLOPs)|11438.0±0.0|285,951.0|
> > > |Data Processed (GB)|0.11719±0.0|2.92975|
> > > |Memory-Time (GB-s)|20.11±0.34|502.75|
> > > |Energy (Joules)|626.67±24.35|15,666.75|
> > > |Wall-Clock Time (s)|2.174±0.037|54.35|
> > >
> > > For all the above different types of metrics, our update policy performs identically, as in the experiments we utilized the same ratio $\frac{\bar \lambda}{\lambda} = \frac{\bar \lambda T}{\lambda T} =0.1$ , which defined the target update frequency. These results demonstrate that our framework is generalizable to different types of resource metrics being used, confirming the proposed method is a general solution for resource-constrained adaptation, not limited to only time as the constraint.
> > >
> > > Q2: We included the computation time comparison with ADWIN and the K-S test to show that traditional drift detection methods add a significant computational overhead from monitoring data streams or performance metrics, as opposed to RCCDA. These are drift detection methods, not decision policies, and are not designed to operate under a resource budget. A direct accuracy comparison of these methods against RCCDA is therefore not applicable. In our paper, we compare RCCDA with other resource-aware policies.
> > >
> > > Q3: Our policy ensures the model stays in a high-performance state as long as possible while meeting resource constraints. In stable, no-drift environments RCCDA will continue the updates until the current loss $f_t$ reaches the best-observed historical loss, at least as low as the pretrained model's initial loss. Once optimal, RCCDA will stop updates, maintaining performance without unnecessary resource use.
> > >
> > > Q4: Thank you for highlighting this. The bound is typically expressed using statistical distances. Papers (R3, R4) put it in terms of KL divergence like ours, while others (R5, R6) assume bounded Total Variation (TV) distance. The two are closely related via Pinsker's inequality, formalizing gradual environmental changes.
> > >
> > > Q5: We included the total memory, decision time, and theoretical complexity for each considered policy in the table below. The window size used for Budget-Increase and -Threshold policies was $w=40$.
> > >
> > > |Policy|Total Memory (bytes)|Time ($\mu$s)|Complexity|
> > > |-|-|-|-|
> > > |RCCDA|276|1.94|$O(1)$|
> > > |Uniform Random|72|1.79|$O(1)$|
> > > |Periodic|108|0.99|$O(1)$|
> > > |Budget-Increase|212+32$w$|1.74|$O(w)$|
> > > |Budget-Threshold|212+32$w$|1.68|$O(1)$|
> > >
> > > Clearly, our policy is comparatively lightweight to the baselines on all evaluated metrics, supporting our claim.
> > >
> > > [R3] Monteleoni, Claire, and Tommi Jaakkola. "Online learning of non-stationary sequences." Advances in Neural Information Processing Systems 16 (2003).
> > >
> > > [R4] Peng, Fu, Meng Zhang, and Ming Tang. "An Information-Theoretic Analysis for Federated Learning under Concept Drift." arXiv preprint arXiv:2506.21036 (2025).
> > >
> > > [R5] Ortner, Ronald, Pratik Gajane, and Peter Auer. "Variational regret bounds for reinforcement learning." Uncertainty in Artificial Intelligence. PMLR, 2020.
> > >
> > > [R6] Zhao, Peng, Long-Fei Li, and Zhi-Hua Zhou. "Dynamic regret of online Markov decision processes." International Conference on Machine Learning. PMLR, 2022.

---

> > > > ### Comment · Reviewer_8AhD · 2025-08-05
> > > >
> > > > Thank you for the additional results. While they support the claim that RCCDA is adaptable to resource-constrained settings, I remain somewhat skeptical about the practicality of the proposed use case. Changes in lighting, seasons, or weather can often be categorized as recurring drifts, which would require RCCDA to perform additional work due to its tendency to "forget" past distributions. Given these considerations, I believe a borderline accept is an appropriate evaluation.

---

> ### Author Response · Authors · 2025-08-08
>
> Thank you for your final comments and engaging with our work. We agree that the challenge of recurring drifts is an important consideration for adaptive systems. As we mentioned before, the key problem is the trade-off between rapid adaptation to the current data distribution and retaining knowledge of past distribution in a model with finite capacity. Our proposed scheme was designed with the primary focus of immediate improvement in the current domain at the cost of generalization to previously seen domains. Although the focus of this work is not in evaluating the generalization capabilities of the models deployed in changing environments, in scenarios with recurring drifts it might be beneficial to preserve the knowledge of past distributions.
>
> To that end, we show in the following that RCCDA can be extended with a replay buffer that retains a small fraction of data from previously seen distributions. When our policy triggers a model update, the training batch can be composed of a majority of new data and a minority of this buffered data. This mechanism helps the model retain information about past concepts, mitigating the effects of catastrophic forgetting.
>
> To demonstrate this, we conducted an experiment comparing two scenarios under a new drift schedule where domains reappear over time. We used the same policy and burst schedules (that went over the photo, art painting, cartoon, and sketch domains in order switching every 100 steps, with the exception of the starting domain being cartoon in the first 50 steps), and modified the data compositions to simulate the following settings:
> - No Buffer: The training dataset shifts entirely to the new domain at predefined intervals.
> - With Buffer: At each shift, the dataset is composed of approximately $80\%$ new data, with the remaining $20\%$ being a balanced split of data from previously seen domains.
>
> The following table presents the model's accuracy on each domain at key time steps, showing the performance just before the next drift occurs.
>
> |Time Step|Domain| No Buffer (100% New Data)|With Buffer (80% New, 20% Old)|
> |-|-|-|-|
> |50|photo|30.60 ± 9.58|30.60 ± 9.58|
> ||art painting|21.35 ± 2.31|21.35 ± 2.31|
> ||cartoon|80.08 ± 11.16|80.08 ± 11.16|
> ||sketch|26.43 ± 3.99|26.43 ± 3.99|
> |150|photo|78.26 ± 3.26|71.74 ± 4.06|
> ||art painting|25.00 ± 1.99|27.60 ± 1.92|
> ||cartoon|20.31 ± 3.04|63.93 ± 2.71|
> ||sketch|18.49 ± 0.80|16.67 ± 6.63|
> |250|photo|50.78 ± 2.76|57.29 ± 4.90|
> ||art painting|63.28 ± 3.31|57.03 ± 3.62|
> ||cartoon|36.33 ± 3.38|60.16 ± 3.76|
> ||sketch|22.01 ± 2.89| 21.88 ± 2.41|
> |350|photo|39.71 ± 4.61|58.07 ± 0.49|
> ||art painting|34.64 ± 3.81|43.75 ± 3.31|
> ||cartoon|70.31 ± 3.68|79.82 ± 5.94|
> ||sketch|30.60 ± 2.89|31.90 ± 1.92|
> |450|photo|12.76 ± 3.41|48.44 ± 7.10|
> ||art painting|15.10 ± 4.47|38.93 ± 6.33|
> ||cartoon|28.39 ± 16.44|70.18 ± 6.20|
> ||sketch|58.07 ± 2.08|57.94 ± 2.96|
> |550|photo|79.43 ± 0.80|79.04 ± 3.10|
> ||art painting|29.43 ± 2.71|41.41 ± 4.82|
> ||cartoon|14.45 ± 6.37|65.10 ± 7.91|
> ||sketch|18.49 ± 0.80|44.66 ± 10.80|
> |650|photo|52.08 ± 2.17|53.52 ± 10.73|
> ||art painting|76.17 ± 10.13|58.59 ± 10.46|
> ||cartoon|38.28 ± 2.73|62.50 ± 6.99|
> ||sketch|29.30 ± 4.07|46.88 ± 6.27|
> |750|photo|42.84 ± 6.95|57.16 ± 6.10|
> ||art painting|37.37 ± 2.31|52.21 ± 6.20|
> ||cartoon|73.83 ± 5.37|75.26 ± 8.60|
> ||sketch|30.86 ± 6.51|47.53 ± 5.12|
> |850|photo|11.72 ± 3.08|55.08 ± 3.08|
> ||art painting|12.24 ± 1.44|39.58 ± 1.76|
> ||cartoon|15.89 ± 5.09|65.62 ± 4.98|
> ||sketch|64.58 ± 7.79|62.50 ± 12.77|
> |950|photo|80.86 ± 4.47|85.29 ± 6.22|
> ||art painting|36.72 ± 1.94|49.35 ± 7.56|
> ||cartoon|23.57 ± 4.31|61.33 ± 9.13|
> ||sketch|18.62 ± 0.97|57.03 ± 7.94|
>
> These results demonstrate several interesting insights. As expected, when seeing the domains for the first time, the original framework is better. However, when a domain occurs again, the buffered approach performs better or comparatively well, as long as the new domain is not too distinct compared to the previously seen ones. In general, we have a trade-off between the immediate model performance in the current domain and model generalization across all the domails. Depending on the application scenario, one can choose either the buffered or non-buffered approach, and our algorithm supports both.
>
> Note that these results are only preliminary. A more detailed investigation into the aforementioned trade-off and use cases of a buffer-based approach would be a fascinating future work direction.

---

### Official Review · Reviewer_EHwV · 2025-07-01

**Clarity:** 3
**Significance:** 2
**Originality:** 2
**Rating:** 4
**Confidence:** 3

**Summary:**

To tackle the high computational overhead inherent in existing concept drift adaptation models, the authors propose a model update policy. The approach specifically applies Lyapunov optimization to minimize the model’s time-averaged expected loss. Theoretical analysis establishes the algorithm’s asymptotic convergence and stability guarantees.

**Questions:**

1. My primary concern is whether the scenario setup is reasonable. Specifically, is there evidence to demonstrate that the computational cost of existing methods for detecting concept drift is unacceptable? The authors need to provide concrete evidence quantifying the computational overhead of existing approaches under specific hardware conditions, and show that this overhead is indeed unacceptable. Otherwise, the paper appears to be more of an exploration under self-defined conditions.

2. How are certain hyperparameters in the algorithm determined? For example, in the threshold-based policy, does \(\lambda\) vary depending on the dataset? I would recommend the authors provide additional details on these aspects to ensure the reproducibility of the algorithm and to verify its generalization capability.

3. The experiments lack validation of the paper's core motivation. The authors should supplement the study with specific results, such as computational parameters and time consumption, to substantiate their claims.

**Ethical Concerns:**

["NO or VERY MINOR ethics concerns only"]

**Final Justification:**

Thank you for the response, i will keep my rating.

**Limitations:**

yes

**Paper Formatting Concerns:**

No Formatting Concerns

**Quality:**

3

**Strengths And Weaknesses:**

Strength
1.Utilizing Lyapunov Optimization is interesting.
2.The paper is well organized

Weakness
1.The application scenario require greater specificity.​

---

> ### Author Rebuttal · Authors · 2025-07-28
>
> We thank the reviewer for valuable feedback and for recognizing the novelty of applying Lyapunov optimization. We believe that our clarifications and new experimental results below address each of the raised concerns.
>
> ## Computational cost of drift detection
> We agree that our paper's core motivation would be strengthened by providing quantitative evidence demonstrating that our derived policy has much lower computational overhead than the existing methods. To address this, during the rebuttal period, **we have conducted a new set of experiments comparing the added computational cost of RCCDA with established drift detection methods** - ADWIN and Kolmogorov–Smirnov (K-S) Test. To detect drift, ADWIN uses sequential data processing and error rate calculation, while K-S test analyzes the statistical properties of incoming data streams. We conducted the new experiment on the PACS dataset with a size of 1024 samples and utilized a single Nvidia H100 GPU. We measured the clock-time taken by each method to finish detecting drift/making a decision after the initial model evaluation. The loss, as well as the ground truth labels and the predicted outputs required by ADWIN, were available to each method at the start of their operation. The results, averaged over 4 domains and 3 seeds each, are as follows:
>
> |Algorithm|Additional Decision Making Time (ms)| Relative Slowdown (vs. RCCDA) |
> |-|-|-|
> |RCCDA|0.000744± 0.000142| 1x|
> |ADWIN|1.845618 ± 0.469732|~2480|
> |KS-Test|125996.5 ± 9605.7| ~169,350,134|
>
> These results provide concrete evidence that the overhead of existing methods may be prohibitive in many practical scenarios. For distribution-based methods like the K-S test, an overhead of approximately 126 seconds on average per monitoring step would be unacceptable for any application requiring real-time monitoring. On the other hand, for the error-based methods represented by ADWIN, the ~2480 slowdown compared to our method represents an unnecessary computational burden added on top of the computation associated with model operation and retraining under strict resource constraints. If compounded over long operation time, it could lead to much faster resource depletion than RCCDA. Additionally, while on the evaluated hardware, a 1.68ms average time is not significant, our work is motivated by resource-constrained scenarios, including on-device ML (e.g. vehicle camera monitoring with limited battery, on-device text prediction, etc) where the computational increase might be exemplified by slower hardware, resulting in the computation time required by ADWIN becoming a challenge. Furthermore, in high-performance real-time scenarios, where every millisecond of compute time is critical and can have severe effects on the operation time and performance of such devices, a lightweight, near-zero overhead policy like RCCDA is a major improvement over the comparable baselines. We will include these results along with the above discussion in the final manuscript of the paper.
>
> ## How are hyperparameters determined
> The parameter $\lambda(t)$ represents the time-varying resource cost of performing a model update. This parameter is **system-defined**, reflecting the real-world cost of an update in a given application, such as energy consumption, wall clock-time, financial cost of a GPU instance, etc. In our experiments, for evaluation simplicity, we set it to a constant value and divide it by $\bar \lambda$, the desired time-average cost. As a result, $\frac{\bar \lambda}{\lambda}$ effectively becomes the desired time-average update frequency of the agent's model. In the main experiment, we consider $\frac{\bar \lambda}{\lambda}= 0.1$, simulating a realistic resource-constrained setting, where the model can only be updated once every 10 evaluation steps. Furthermore, in the supplemental material section, we evaluate different desired update frequencies on a single drift schedule to formalize the effect of policy given different degrees of resource availability. Other tunable hyperparameters related to the decision process and the network updates were **determined experimentally via grid search**.
>
> All utilized hyperparameter values are available in the "Supplemental Material: Detailed Experimental Setup, Documentation" section 1.3 and 1.4. These sections demonstrate that the proposed policy can be tuned to generalize across a variety of datasets and dynamic environments. Additionally, we included the code used to conduct our experiments in the supplemental materials, including the different settings testing a variety of hyperparameter values, which improves the reproducibility of the proposed algorithm and our experimental results.
>
> ## Validation of paper's core motivation
> We thank the reviewer for this important comment. The **additional experiment comparing RCCDA with the existing methods demonstrates the lightweight nature of the proposed algorithm**. On the other hand, the effectiveness of our proposed policy is supported by the results in Table 1 and Figure 2, demonstrating that compared to similar policies that have to adhere to resource constraints, our policy achieves robust performance, outperforming the existing baselines on a variety of drift schedules and across the tested datasets. Finally, Figure 2 demonstrates that the update rate of our frequency always tends to the desired set update frequency, validating Theorem's 5.3 claim, and demonstrating adherence to resource constraints. Thorough these experimental results and our theoretical analysis, **we validate the core paper's motivation** stated in our research question: "Can we develop a lightweight, resource-aware policy for dynamic model updates that (i) effectively mitigates performance loss incurred by concept drift in real-time applications while (ii) theoretically guaranteeing adherence to strict resource constraints?" In the final version of the manuscript, we will include the new experiment comparing the computational overhead with the established drift-detection algorithms as well.
>
> Once again, thank you for your effort in reviewing our paper and providing important comments. If there are additional questions we could answer or issues we could answer we would be grateful for an opportunity for further discussion.

---

> > ### Comment · Reviewer_EHwV · 2025-08-05
> >
> > Thank you very much for the response. However, i still hope the authors will give more consideration to Reviewer 8AhD's comments and add the experiments in repsonse to the manuscript. And I will keep my rating, borderline accept.

---

> > > ### Author Response · Authors · 2025-08-08
> > >
> > > Thank you for the additional feedback. With our latest response to the reviewer 8AhD, we have considered all the comments the reviewer raised. We will include the additional results in the final version of the manuscript.

---

### Official Review · Reviewer_3pwb · 2025-07-01

**Clarity:** 3
**Significance:** 2
**Originality:** 3
**Rating:** 4
**Confidence:** 4

**Summary:**

This paper addresses the challenge of adapting machine learning models to concept drift in resource-constrained environments. The authors propose techniques that dynamically select an appropriate retraining policy.

**Questions:**

Why are the bounds stated for \theta_t, D_{t+1} while the objective in equation (5) is defined for \theta_t, D_t?

What is the relationship between the proposed bounds and those in the single-task setting? As I understand, in the single-task case (T = 1), generalization bounds are typically on the order of 1/ \sqrt{n}, where n is the number of samples. The connection between the proposed bounds and the number of samples per dataset is unclear and would benefit from further explanation.

What is the relationship between the proposed bounds and existing state-of-the-art bounds?

**Ethical Concerns:**

["NO or VERY MINOR ethics concerns only"]

**Final Justification:**

The authors have addressed all of my questions and comments. In addition, the paper provides theoretical guarantees in settings with concept drift, which are particularly challenging.

**Limitations:**

The paper discuss some limitations. I also believe that the assumptions made in the paper represent a limitation of the proposed approach.

**Quality:**

2

**Strengths And Weaknesses:**

Strengths:

The paper addresses the problem of learning under concept drift.

The paper provides theoretical guarantees in settings with concept drift.

Weaknesses:

Are the assumptions made in the paper standard? It seems like a strong assumption to require some form of prior knowledge and access to a loss estimator.

Do [10] and [26] refer to the same work?

The proposed problem is particularly interesting in scenarios where the dataset is small. If the number of samples per dataset is large, one could simply learn using only the data from that sample set, making the concept drift less critical. However, in the small-sample regime, is it still reasonable to assume access to an unbiased estimate of the loss?

---

> ### Author Rebuttal · Authors · 2025-07-29
>
> We thank the reviewer for their insightful feedback. We appreciate that the reviewer recognized the value of providing theoretical guarantees for learning under concept drift. Below we further address the raised concerns and questions.
>
> ## Assumptions in the paper.
> We would like to clarify a few points on "**prior knowledge**" with respect to our assumptions. The L-smoothness, bounded variance, and bounded loss assumptions are **standard assumptions** common in most theoretical analyses of stochastic optimization algorithms. Furthermore, the assumption about bounded KL-divergence between two time steps reflects the fact that the data distribution will not change arbitrarily quickly in practice. We note that even though we make this assumption, we only utilize it to provide convergence and time-average constraint violation bounds; RCCDA does not use any prior knowledge of concept drift throughout its operation.
>
> When it comes to the assumption of the estimate of the loss, our method uses the model's inference loss, calculated when ground truth labels are available. This is a **foundational assumption** for many of the most-cited methods in the concept drift literature. For instance, the widely used drift detectors we mentioned in the related works section, like ADWIN and EDDM operate by monitoring the model's error rate, which explicitly requires access to ground truth labels. While we do not explicitly try to detect or categorize the drift in our paper, assuming loss availability aligns with the standard requirements for supervised drift adaptation. Additionally, this assumption corresponds to a practical setup for many real-world systems like text or ad-click prediction, trend-tracking, or content moderation in which real-time feedback is available.
>
> ## Duplicate citation ([10] and [26])
> We are grateful to the reviewer for pointing out this issue. The mentioned references are indeed the same work, and we will correct this citation error of the final version.
>
> ## Assumption on access to an unbiased estimate of the loss for small datasets
> This assumption stems directly from being able to access the model's inference loss at any time $t$, and as mentioned beforehand, it is a **standard assumption in supervised drift adaptation literature**. Regarding its feasibility, consider that for a larger dataset, small parts of it may be labeled, resulting in a good estimate of the inference loss. If $\mathcal D_t$ is small, the empirical loss can result in a high-variance estimate of the true loss. However, even though the variance might increase, even for small datasets, as long as it is possible to sample a batch $\xi_t$ representative of the distribution at time $t$, the empirical loss should not include any bias for that dataset.
>
> To address the issue with variance in the estimate of the loss, we conducted an **additional experiment evaluating how adding error to the loss estimate affects the agent model's performance**. The experiment was conducted on the PACS dataset, using the "burst" drift schedule. In the experiment, once the evaluation loss was calculated, a scaled value sampled from a normal distribution with mean 0, and varying degrees of variance was added to the original loss value. Mathematically, the uncertain loss is modeled as $f_t^{\text{estimate}} = f_t + \mathcal N(0, (k_u \cdot f_t)^2)$, where $k_u$ is the degree of uncertainty. The results of this experiment are available in the table below:
>
> |Degree of uncertainty $k_u$ | Average Validation Accuracy (%) | Average Update Rate $\frac{\bar\lambda}{\lambda}$ | ($K_p$, $K_d$) Hyperparameters|
> |-|-|-|-|
> |0 (certain result) |79.4 ± 5.2| 0.093 ± 0.030| (1.5, 0.3) |
> |0.05 |79.3 ± 6.1|0.099 ± 0.011 | (0.7, 0.3) |
> |0.1 |79.2 ± 6.8|0.103 ± 0.009| (0.6, 0.25) |
> |0.2 |78.0 ± 7.0|0.103 ± 0.007| (0.4, 0.25) |
> |0.3 |77.4 ± 8.0| 0.104 ± 0.004| (0.3, 0.25) |
>
> Naturally, with growing estimation uncertainty, we observe some performance degradation at the same update rates. Nonetheless, overall, RCCDA demonstrates a high degree of robustness to loss estimation error. The largest performance drop in validation accuracy is only about 2.5% relative to the case when the exact value of the loss is known, achieved when the loss uncertainty becomes fairly high, with an error variance equal to 30% of the loss at any given time $t$. We also observe an increase in the standard deviation of the average validation accuracy as the uncertainty grows, reflecting higher variability in model performance due to the added estimation noise. Despite that, the new result demonstrates that the proposed algorithm performs robustly even when the loss estimate is noisy. This is particularly relevant for small-dataset regimes, where inherent sampling variance would have a similar effect on the estimate of the loss.
>
> We will add this experiment and associated discussion to the revised manuscript.
>
> ## Response to questions
> **Why are the bounds stated for $\theta_t, D_{t+1}$ while the objective in equation (5) is defined for $\theta_t, D_t$**: This is a direct result of the **proof technique used for online optimization**, available in appendix A. The objective is to minimize the cumulative loss over time under some resource constraint. To analyze this, we derive convergence bounds by examining the system's evolution from one discrete time step $t$ to the next. The final convergence bound arises from conducting that analysis and utilizing various techniques to provide a bound on the gradients. The Lyapunov optimization objective comes from attempting to minimize the derived convergence bound.
>
> **Single-task setting bounds relationship**: Standard generalization bounds, typically on the order of $\mathcal O(1/\sqrt{n})$, measure how well a model trained on $n$ samples from a static distribution will perform on new data from that **same distribution**. In contrast, our work provides a convergence bound on the expected squared magnitude of the gradient for a dynamic setting with concept drift. This bound addresses a fundamentally different challenge of characterizing the model's performance over a time horizon $T$ as the **underlying data distribution $p_t$ evolves**. Because we consider non-stationary setting, where the distribution of the data changes over time, deriving a traditional generalization bound is highly challenging, as the standard assumption of a fixed data distribution becomes inapplicable to the considered scenario.
>
> **Relationship to existing state-of-the-art bounds**: As stated in the paper, to the best of our knowledge, **no prior work provides convergence and stability guarantees for model adaptation under concept drift subject to a strict, time-averaged resource constraint**. Existing works on concept drift primarily focus on either detecting drift or adapting to it assuming unconstrained updates. The bounds provided in these works do not account for a restrictive resource budget that may prevent the model from updating at the theoretically optimal moments.
>
> Thank you for your insights and effort in reviewing our paper! We will add all of these clarifications to the manuscript. If there are other questions we could answer or issues we could address, please let us know.

---

> ### Comment · Reviewer_3pwb · 2025-08-02
>
> The authors have addressed all of my questions and comments. In addition, the paper provides theoretical guarantees in settings with concept drift, which are particularly challenging. Therefore, I have decided to raise my score to a weak accept.

---

### Official Review · Reviewer_bCey · 2025-07-02

**Clarity:** 3
**Significance:** 3
**Originality:** 3
**Rating:** 5
**Confidence:** 3

**Summary:**

In this paper the authors introduce RCCDA, an adaptive model update policy designed to address concept drift in machine learning under strict resource constraints. The policy leverages Lyapunov analysis to minimize a drift-plus-penalty term, ensuring stability and adherence to resource constraints. The authors showed theoretical convergence bounds. They also demonstrated the policy's effectiveness through experiments on domain generalization datasets. RCCDA demonstrates superior accuracy and drift recovery compared to baseline policies, while efficiently utilizing resources.

**Questions:**

How does RCCDA handle scenarios where the estimation function for gradient magnitude is inaccurate or noisy?

Can RCCDA be extended to handle multi-modal concept drift or non-stationary environments?

What are the computational overheads of maintaining the virtual queue?

It would be good to Include some benchmarks against state-of-the-art adaptive policies to highlight comparative advantages.

**Ethical Concerns:**

["NO or VERY MINOR ethics concerns only"]

**Final Justification:**

The authors have answered all my questions and have agreed to add the societal impact of their work to the main paper. I think this is a good paper and can be accepted.

**Limitations:**

The authors have explicitly discussed the limitations of their work, which is commendable. However they have not discussed any societal impact. It would have been good if they had included a discussion on broader implications (e.g., fairness in resource allocation or unintended consequences in deployment) in the paper.

**Quality:**

3

**Strengths And Weaknesses:**

The paper provides a robust theoretical foundation for the RCCDA policy. Experimental results seem to be reproducible, with detailed hyperparameters and settings provided. The results also seemed to be quite significant.

Moreover the paper is well-structured, with clear explanations of the theoretical framework and experimental setup. Limitations are explicitly discussed, which is laudable.

The focus on concept drift in resource-constrained environments is a real world problem which needs enough attention.

 RCCDA introduces a Lyapunov drift-plus-penalty framework, ensuring stability and adherence to resource constraints. It dynamically adjusts update timing based on historical loss data, outperforming static or heuristic policies. These contributions seem to be unique and novel.


The paper is really well written. My only concern is its assumptions about loss availability which may not be feasible in real world scenarios.

---

> ### Author Rebuttal · Authors · 2025-07-29
>
> We thank the reviewer for their thorough and positive assessment. We are grateful that they found our paper's theoretical foundation robust, the results significant, and the contributions novel. Below we address the raised concern and answer the posed questions.
>
> ## Loss availability
> Our assumption is a standard one used in **supervised drift detection** methods. Establish methods like ADWIN and DDM, which operate on the error rate, also require ground truth labels in practice. We can here point to scenarios where the labeled data for loss calculation becomes available right away or with a slight delay, e.g., in tasks such as text prediction (predicted and ground truth text are both available in real time), content moderation (labels are provided through feedback), and online behavior tracking (real-time user responses).
>
> The alternative is **unsupervised drift detection**, which monitors feature distributions and therefore does not require labels. However, these methods are computationally intensive. To see this, we have conducted a new set of experiments comparing the added computational cost of RCCDA with the cost of running (supervised) drift detection using ADWIN, as well as detecting drift using the (unsupervised) Kolmogorov–Smirnov (K-S) test, which analyzes the incoming data stream to detect drift. The experiments were conducted on the PACS dataset with a size of 1024 samples using a single Nvidia H100 GPU. We measured the wall clock-time taken by each method to finish detecting drift/making a decision after the initial model evaluation. The results, averaged over 4 starting domains and 3 seeds each are as follows:
>
> |Algorithm|Additional Decision Making Time (ms)| Relative Slowdown (vs. RCCDA) |
> |-|-|-|
> |RCCDA|0.000744± 0.000142| 1x|
> |ADWIN|1.845618 ± 0.469732|~2480|
> |KS-Test|125996.5 ± 9605.7| ~169,350,134|
>
> The decision time of approximately two minutes for the K-S test would be prohibitive for real-time monitoring applications, making such methods impractical for the evaluated setting. Additionally, these results demonstrate that given the same assumptions, by using just the loss value for the policy decision, we can achieve significant speedup compared to detecting the drift using error rate calculation, which is the method utilized by ADWIN. We will include these results and this discussion in the final version of the paper.
>
> ## Response to questions
> **How does RCCDA handle inaccurate/noisy gradient estimation**: Our policy is designed around the Lyapunov optimization framework's virtual queue, which acts as a long-term controller. This ensures that **regardless of the noise in the gradient estimation term, the long-term time-average resource constraint will be satisfied**, as established by our stability analysis in Theorem 5.3. However, noisy estimates might lead to suboptimal retraining decisions at certain time steps; depending on how noisy the estimate is, the inference performance of the agent's model might deteriorate over time. Note that in our experiments, we utilize an estimation function of $\hat{g}(\cdots) = K_{d} (f_t-f_{t-1})$, which is a simple and lightweight approach to estimating the gradient magnitude with which we are able to achieve robust performance across the variety of tested datasets and drift schedules.
>
> **Can RCCDA be extended to multi-modal drift**: As we utilize only loss and gradient magnitude in decision making, **the theoretical analysis is generalizable to a multi-modal setting**, supporting different modalities separately as well as potentially multi-modal tasks. Due to computational constraints, we only considered single-modal concept drift in the paper. When it comes to the evaluated environments, simulating the concept drift through domain changes gives us a convenient mechanism for evaluating non-stationary environments. This is also encapsulated in the theoretical analysis by the drift-induced loss term $\Delta f_{\delta}(t)$ in Theorem 5.1 and then the drift bound $\delta(t)$ in Corollary 5.2.
>
> **Computational overhead of maintaining the virtual queue**: Maintaining the virtual queue adds **negligible overhead**. The queue itself keeps track of only a single floating point number. The update at each step involves one addition, one subtraction, and one $\max$ operation, resulting in $\mathcal O(1)$ time complexity in both time and memory, making the added overhead minimal from a resource perspective.
>
> **Benchmarks against SOTA adaptive policies**: In the earlier response we have included a new comparison of times required to execute operations required to make a policy decision. In our main paper, we further included comparisons with baseline policies that guarantee adherence to strict resource constraints but are not designed with a learning objective in mind. Other retraining policies do not explicitly guarantee they will provide the desired long-term time-averaged resource utilization, and often don't even consider operating under limited resource budget [10, 11, 12, 22, 27, 28]. As such, to the best of our knowledge, the evaluated baselines, as well as the added computational complexity comparison, are a reasonable set of benchmarks to compare against. Through our experimental section we demonstrate that our policy is capable of outperforming these benchmarks in terms of recovery time from drifts under a fixed resource budget.
>
> ## Societal impact discussion
> This is an excellent point. RCCDA has a variety of broader positive impacts. The reduced computational load on devices coupled with maintaining overall high performance results in reduced computational load of devices, which then reduces energy consumption and lowers operational costs for individuals or organizations running models on the edge. Furthermore, RCCDA's lightweight nature makes it feasible to run on low-cost low-power hardware, allowing advanced AI capabilities to be deployed in a wider range of settings. Furthermore, by enabling on-device models to adapt to concept drifts, RCCDA allows for better personalization over time while preserving computational resources.
>
> At the same time, we acknowledge that there might be some negative impacts associated with RCCDA as well. For example, if the incoming data streams become biased, our policy might update the model to fit the new biased data, potentially reinforcing negative sentiments. In-line with this, we did not consider adversarial settings where some malicious entity might inject the agent's data stream with malicious data, causing our policy to decide to update the model, degrading its performance. We will add the above discussion to the final version of the paper.
>
> Once again, we would like to thank you for your time and effort invested in reviewing our paper. If you have any further additional questions or concerns please let us know, we will gladly address them.

---

> > ### Comment · Reviewer_bCey · 2025-08-05
> > **Thanks for the response**
> >
> > Thanks for the response and additional results. I found them adequate, and I have updated my reviews to reflect that.

---

### Official Review · Reviewer_iUQT · 2025-07-03

**Clarity:** 2
**Significance:** 3
**Originality:** 3
**Rating:** 4
**Confidence:** 3

**Summary:**

This paper presents an adaptive threshold-based policy that determines the optimal time
to update the parameters of a predictive model in the presence of concept drift, using only past loss information, and ensuring strict bounds on constraint violations. The proposed policy triggers updates based on accumulated loss and a virtual queue tracking resource expenditure.
The paper also presents a convergence analysis on the relationship between model accuracy and computational cost by estimating bounds on the model accuracy depending on the update policy and the concept drift.
Finally, the paper shows empirically that the proposed solution is able to outperform common baselines for concept drift on three datasets for domain generalization.

**Questions:**

- Is there a reason to limit the evaluation to vision? Other modalities (e.g. audio or text) could also be evaluated.

- What kind of recognition model is used? The theoretical analysis is independent of the model. However, the empirical evaluation should be tested on different models.

**Ethical Concerns:**

["NO or VERY MINOR ethics concerns only"]

**Final Justification:**

The authors answered all my questions and clarified my doubts with additional experiments with new datasets and new domains. I confirm my positive evaluation of the paper.

**Limitations:**

yes

**Paper Formatting Concerns:**

No formatting contraints

**Quality:**

3

**Strengths And Weaknesses:**

\+ The proposed algorithm has some guarantees on the quality of the solution while keeping a constrained resource consumption.

\+ The paper is well presented even though the figures could be more clear and there a some typos in the text.

\- It is not clear what is the recognition model used for the experiments. Different models with different capacity should be evaluated. Also, the proposed approach could be applied to different domains than vision. Is there a specific reason to limit experiment to visual recognition?

\- The empirical evaluation is limited to three datasets. To better verify the quality of the proposed solution, authors could consider other datasets with multiple domains such as TerraIncognita and DomainNet.

---

> ### Author Rebuttal · Authors · 2025-07-30
>
> We thank the reviewer for the constructive feedback and the opportunity to clarify our experimental design and further validate the generalizability of our approach. We address the points regarding model diversity and evaluation modalities below.
>
> ## Different capacity models
> We acknowledge the reviewer's point about the importance of evaluating our method on models with different capacities. We would like to clarify that we had incorporated this in our experiments by using **three distinct neural network architectures** across the three vision datasets, each with varying complexity:
> - For the PACS dataset, we used a custom deep convolutional neural network with residual connections to handle the significant domain shift between different image styles. The model had 11,177,223 parameters in total, all trainable.
> - For the DigitsDG dataset, we utilized a more standard, shallower convolutional neural network suitable for the simpler task of digit recognition. The model had 1,907,146 parameters in total, all trainable.
> - For the OfficeHome dataset, we adapted a pretrained ResNet-18 model to handle the higher complexity of more classes and larger images. During both pretraining and evaluation, we froze the initial layers and exclusively fine-tuned the final convolutional block (layer4) and our custom classification head. The model had 11,324,545 parameters in total, of which 8,541,761 were trainable.
>
> Testing across different model sizes and architectures allowed us to confirm that **RCCDA's performance is robust across varying model capacities**. This aligns with our theoretical analysis, which is model-agnostic as long as the loss function satisfies the stated standard assumptions.  The complete architectural specifications for each model, along with all training and evaluation hyperparameters, are provided in the "Supplemental Material: Detailed Experimental Setup, Documentation" for full transparency and reproducibility.
>
> ## Different modalities
> Our theoretical analysis is domain-agnostic, i.e., the provided bounds are applicable to all domains. We utilized the **vision domain in our experiments as it allowed us to model the desired environment in a straightforward way** and demonstrate the efficacy of RCCDA. The domain generalization datasets readily emulate concept drift, as pictures in different domains will have different underlying distributions. Furthermore, the vision models pre-trained on only a single domain performed poorly on new domains, which demonstrated the need for model updates, and allowed us to test our proposed algorithm in settings it would likely be deployed.
>
> When it comes to utilizing **other modalities, the experimental setup becomes more challenging**, and simulating concept drift becomes less straightforward. To demonstrate this, we added an experiment on the MEMD-ABSA dataset [R1] - a multidomain dataset developed for textual Aspect-Based Sentiment Analysis (ABSA) task. The dataset contains reviews partitioned into five distinct domains: books, clothing, hotel, laptop, and restaurant. Similar to our main experiments, we first pre-trained a neural network - the TinyBERT [R2] model - on each domain independently. Then, each pre-trained model was deployed in an environment simulating the 'burst' drift schedule, where the domain of the incoming text data changes abruptly at specific intervals, simulating sudden shifts in the underlying distributions. However, the difference in the domains was not substantial enough to be able to simulate a meaningful concept drift well. We illustrate this in the table below, demonstrating the testing accuracy of a model pre-trained on one domain when evaluated on all other domains:
>
> |Pretrained \Tested | Books | Clothing | Hotel | Laptop | Restaurant |
> |-|-|-|-|-|-|
> |Books|81.7% ± 1.7%|80.3% ± 0.6%|88.2% ± 1.8%|73.4% ± 1.6%|73.6% ± 2.9%|
> |Clothing|75.0% ± 0.3%|84.8% ± 0.6%|88.8% ± 1.2%|74.8% ± 1.6%|74.5% ± 0.2%|
> |Hotel|72.0% ± 1.1%|77.5% ± 0.9%|95.8% ± 0.4%|66.1% ± 1.8%|72.6% ± 1.5%|
> |Laptop|70.7% ± 1.5%|80.9% ± 1.2%|84.2% ± 3.3%|80.1% ± 0.2%|73.8% ± 1.5%|
> |Restaurant|78.1% ± 1.4%|81.3% ± 0.5%|91.4% ± 1.4%|75.6% ± 1.0%|82.2% ± 0.3%|
>
> As demonstrated by these results, while the model performance drops when evaluated on unseen domains, the pre-trained model still remains fairly competitive achieving relatively high classification accuracy. This is in contrast to the vision-based tasks where different domains caused a substantial performance drop, as can be seen in Figure 2 in the main paper, where the validation accuracy drops from around 90% to around 35% the moment a new domain is introduced to the dataset. In other words, for certain tasks, the **drift associated with domain changes is small**. As a result, when we evaluated RCCDA and the baselines in the ABSA setting, the performance improvement was less significant compared to other datasets. The table summarizing the results for the "burst" drift schedule is available below, with an effective update rate $\frac{\bar \lambda}{\lambda}=0.01$.
>
> |Policy | Average Validation Accuracy (%) |
> |-|-|
> |RCCDA (Ours) | **81.4 ± 2.9** |
> |Uniform| 79.8 ± 3.5|
> |Periodic|80.0 ± 3.2|
> |Budget-Increase|79.2 ± 3.6|
> |Budget-Threshold|79.2 ± 4.2|
>
> Of note here is the fact that some of the baselines achieved the accuracy of approximately 80% with almost no updates - having effective average update rate of 0.001. This demonstrates that due to the small magnitude of the drift, the performance boost achieved by our method is not as significant as the performance gain achieved by model's generalization ability.
>
>
> ## Additional dataset
> To further validate our proposed solution, we conducted an additional experiment on the DomainNet dataset. Due to time constraints during the rebuttal period, we restricted the experiments to only 4 domains of the DomainNet dataset: real, clipart, painting, and sketch. We pre-trained a modified version of the ResNet-50 model on each of these domains, and deployed the pre-trained model in an environment experiencing concept drift simulated by the "burst" drift schedule, abruptly injecting new domain data at specific intervals. The effective update rate $\frac{\bar \lambda}{\lambda}$ was set to $0.1$, just as in the main paper experimental setup. After tuning the hyperparameters and evaluating the performance of the baselines, we arrived at the results summarized in the table below:
>
> |Policy | Average Validation Accuracy (%) |
> |-|-|
> |RCCDA (Ours) | **68.9 ± 2.1** |
> |Uniform| 66.6 ± 3.4|
> |Periodic|67.8 ± 4.3|
> |Budget-Increase|63.7 ± 4.0|
> |Budget-Threshold|65.6 ± 2.2|
>
> Compared to the other policies, when averaged over all seeds and domains, our policy once again achieved highest average validation accuracy, further corroborating the findings from PACS, DigitsDG, and OfficeHome from the main paper.
>
> We sincerely thank the reviewer for their careful and thorough review of our work. We will gladly address any additional questions or concerns.
>
> References:
>
> [R1] Cai, Hongjie, et al. "Memd-absa: A multi-element multi-domain dataset for aspect-based sentiment analysis." Language Resources and Evaluation (2025): 1-29.
>
> [R2] Jiao, Xiaoqi, et al. "Tinybert: Distilling bert for natural language understanding." arXiv preprint arXiv:1909.10351 (2019).

---

> > ### Comment · Reviewer_iUQT · 2025-08-04
> >
> > I thank authors for their answers which clarified my doubts with additional experiments with new datasets and new domains.
> > I confirm my positive evaluation of the paper.

---

### Note · Authors · 2025-08-13

We thank the reviewers for their insightful feedback, which helped us improve the depth and clarity of our work.

We believe that through our discussions, all major concerns have been addressed, further highlighting the contributions of our work. Our paper introduced RCCDA, a new adaptive policy that addresses the long-standing open challenge of maintaining model performance under concept drift while adhering to strict resource constraints.

In response to the reviewers' feedback, we have incorporated the following new experiments and clarifications:
- **Quantifying resource usage**: We have provided new experimental results demonstrating that our framework is generalizable to different types of resource constraints, such as computation (GFLOPs), energy (Joules), or time (s).
- **Lightweight approach**: We added experiments comparing the computational overhead of RCCDA with the established drift detection methods, ADWIN and the Kolmogorov-Smirnov (K-S) test, and compared the memory usage, execution time, and time complexity of RCCDA with the evaluated baselines. The results supported our claims that RCCDA is a lightweight policy, comparable in memory and time usage to the baselines, and is much faster than the existing drift-detection-based methods, which highlighted the need for a new policy.
- **Modalities Diversity**: We clarified that our analysis is generalizable to different types of modalities, and added a new experiment on a textual dataset (MEMD-ABSA) to demonstrate it. We further clarified that our original experiments utilized different model capacities and added a new experimental result on a fourth image dataset to confirm the effectiveness of our approach.
- **Recurring drifts handling**: We acknowledged the importance of recurring drifts and demonstrated that while the evaluated setting is a different problem from continual learning, RCCDA can be extended to handle these cases better with a replay buffer mechanism. Preliminary experiments demonstrated that this approach could mitigate the problem of catastrophic forgetting when domains reappear, resulting in a trade-off between current real-time and generalization performance.

We believe the additional results and clarifications strongly support our paper's main contributions and demonstrate RCCDA's efficiency, generality, and theoretical soundness as a unique solution for real-time ML deployments.

---

### Decision · Program_Chairs · 2025-09-17

**Decision:**

Accept (poster)

**Comment:**

This work studies the problem of adapting model training when the data distribution is drifting over time under while constraining the number of times the model can be updated. The authors show that the proposed algorithm, RCCDA, will satisfy some notion of first-order optimality for the time-averaged gradients, that depends on the amount of concept drift, while approximately satisfying the resource constraints. The theoretical guarantees are accompanied by an empirical evaluation.

All reviewers agree that this is an important problem with real world applications, that the paper is well-written and that the theoretical findings are somewhat interesting. The main concerns of reviewers were about the practicality of the algorithm and the somewhat limited empirical evaluation. The authors address these two points by running extra experiments and a careful discussion on the computational complexity, together with a table that verifies the efficiency of the proposed algorithm.

Overall the paper seems to provide an interesting and novel approach for solving the concept drift problem that comes with theoretical guarantees and a good empirical evaluation.